# Contrast-enhanced ultrasound measurement of pancreatic blood flow dynamics predicts type 1 diabetes progression in preclinical models

Joshua R. St Clair[1], David Ramirez[1], Samantha Passman[1] & Richard K.P. Benninger[1,2]

In type 1 diabetes (T1D), immune-cell infiltration into the islets of Langerhans (insulitis) and β-cell decline occurs many years before diabetes clinically presents. Non-invasively detecting insulitis and β-cell decline would allow the diagnosis of eventual diabetes, and provide a means to monitor therapeutic intervention. However, there is a lack of validated clinical approaches for specifically and non-invasively imaging disease progression leading to T1D. Islets have a denser microvasculature that reorganizes during diabetes. Here we apply contrast-enhanced ultrasound measurements of pancreatic blood-flow dynamics to non-invasively and predictively assess disease progression in T1D pre-clinical models. STZ-treated mice, NOD mice, and adoptive-transfer mice demonstrate altered islet blood-flow dynamics prior to diabetes onset, consistent with islet microvasculature reorganization. These assessments predict both time to diabetes onset and future responders to antiCD4-mediated disease prevention. Thus contrast-enhanced ultrasound measurements of pancreas blood-flow dynamics may provide a clinically deployable predictive marker for disease progression in pre-symptomatic T1D and therapeutic reversal.

[1] Department of Bioengineering, University of Colorado Anschutz Medical Campus, Aurora, 80045 CO, USA. [2] Barbara Davis Center for Diabetes, University of Colorado Anschutz Medical Campus, Aurora, 80045 CO, USA. Correspondence and requests for materials should be addressed to R.K.P.B. (email: richard.benninger@ucdenver.edu)

Type 1 diabetes (T1D) involves infiltration of auto-reactive T-cells into the pancreatic islets (insulitis), local inflammation, and destruction of insulin-producing β-cells[1,2]. For most patients with T1D, there is a pre-clinical asymptomatic ("pre-symptomatic"[3]) phase several years prior to clinical presentation of diabetes, where insulitis and immunological abnormalities are present but β-cell mass and glucose-stimulated insulin secretion remains at significant levels to regulate blood glucose and for the disease to remain asymptomatic. This pre-symptomatic period presents an opportunity for therapeutic intervention to preserve the β-cell mass and ultimately prevent disease progression[4,5]. At disease onset most, but not all of the β-cells have been destroyed[6], suggesting that therapeutic intervention may be more successful when applied prior to T1D onset[5,7–9]. The presence of multiple islet-associated auto-antibodies can predict the disease onset[10], but they are not pathogenic and do not reflect a therapeutic potential to reverse the underlying disease. While approaches examining islet microvascular permeability holds promise[11,12] to date, there is an absence of approaches that can monitor the underlying insulitis and the decline in total β-cell mass associated with the pre-symptomatic phase of T1D. This presents a major barrier for implementing preventative therapies for patients at risk for T1D, or who have recently developed T1D.

To date, potential therapeutic interventions have also had limited success in preserving β-cell mass or c-peptide response. For example, antiCD3 immunotherapy provides c-peptide preservation for 1–2 years only[13,14]. However, preservation has been demonstrated to be highly heterogeneous where some patients show no preservation, but others show extensive preservation: so called "non-responders" and "responders", respectively[15]. Therefore, developing techniques to monitor the halting of underlying disease progression will also enable monitoring the efficacy of preventative treatments.

Approaches to specifically label β-cell mass in vivo have proven problematic, potentially due to the exceptionally low number these cells represent, relative to the overall pancreas volume (~1%). Importantly, pancreatic islets are highly vascularized and receive approximately 10–20% of pancreatic blood volume, despite representing only 1–2% of the pancreatic mass[16]. Changes in islet microvasculature can occur in response to acute glycemic changes and can also occur pathophysiologically during diabetes[16–20]. For example, streptozotocin (STZ) treatment results in inflammation-induced increases in islet microvascular permeability and dilation[21–23]. Changes in the islet microvasculature also occurs in non-obese diabetic (NOD) mice[24,25], which correlates with the observed changes in islet blood flow[18,20]. Furthermore, decreases in islet vascular density and compensatory increases in vessel diameter have been observed in mouse models of type 2 diabetes (T2D)[26] and human T2D donors[27]. Given that a disproportionate percentage of pancreatic blood is delivered to the islets, microsphere deposition experiments have established that the total pancreatic blood flow is highly correlated with the islet blood flow[20]. Given the changes to islet microvasculature and islet blood flow with diabetes, we anticipated that real-time measures of the pancreatic blood flow and the changes in these measures over time would provide a non-invasive way to diagnose and follow the state of the islet through the progression of T1D.

In this study, we sought to develop, verify, and utilize contrast-enhanced ultrasound (CEUS) as a means for measuring pancreatic blood flow dynamics as a predictive marker of T1D initiation, disease progression, and disease reversal following immunomodulatory therapy. CEUS has been previously established to measure and quantify blood flow dynamics in the kidneys[28,29], heart[30], brain[31,32], and the pancreas during pancreatitis[33] and pancreatic tumors[34,35]. Importantly, micro-bubble contrast agents are FDA approved for echocardiography approaches (e.g., DEFINITY, Lantheus) and liver applications including pediatric populations (LUMASON, Bracco). CEUS provides several advantages over positron-emission tomography (PET) or magnetic resonance imaging (MRI) that includes lack of harmful exposure to ionizing radiation, no side effects on the thyroid or kidney, and an exceptional ease of usage and deployment[36]. To our knowledge, CEUS has not been used for imaging of pancreatic changes associated with diabetes progression and diabetes reversal. Such a tool could provide a completely novel in vivo imaging paradigm for the identification, diagnosis, and monitoring of diabetes progression in patients. Here, we measured pancreatic blood flow dynamics in several pre-clinical models of T1D, and correlated the changes in blood flow dynamics with histological analyses, disease onset, and the prediction of successful therapeutic interventions.

## Results

**Measuring islet blood-flow dynamics by contrast enhanced ultrasound.** To provide information regarding variations in islet blood flow and, in turn, variations in islet microvasculature organization, we performed high-frequency CEUS measurements in mice receiving a bolus of 3–4 μm size-isolated gas-filled lipid microbubbles (SIMBs, Advanced Microbubbles Laboratories, Boulder CO[37]) via a tail-vein catheter. SIMB delivery resulted in an elevated non-linear (NL) contrast signal reaching a steady-state after 10–20 s (Fig. 1a, b). In order to consistently measure the blood flow dynamics independent of SIMB concentration and bubble degradation, a high mechanical-index pulse ("flash destruction") was applied within 30 s of SIMB bolus delivery, which destroyed SIMBs within the plane of the transducer and reduced NL contrast by 69 ± 2% ($n = 134$ scans). Following SIMB flash destruction, we measured the kinetics of NL contrast recovery within the pancreas resulting from SIMB replenishment (Fig. 1a–c). The rate of this NL contrast "reperfusion" reflects changes in the velocity of pancreatic blood perfusion (Fig. 1c)[31], whereas the reperfusion amplitude reflects the volume of filling by blood perfusion (Fig. 1c)[31]. We observed some variability within the groups of animals used (Supplementary Fig. 1A-C, for C57BL6 SD/mean reperfusion rate ~70%, amplitude ~43%), with daily consecutive reperfusion measurements within mice being consistent with this variability (Supplementary Fig. 1D, mean absolute change reperfusion rate ~78%, amplitude ~45%), and hourly consecutive reperfusion measurements within mice being slightly below this variability (Supplementary Fig. 1E, mean absolute change reperfusion rate ~69%, amplitude ~32%). There was no association of reperfusion rate with any minor variations in blood-glucose levels under anesthesia (Supplementary Fig. 1F–I). Across all animals within this study, there were no significant correlations of reperfusion rate with weight, respiration rate, or body temperature (Supplementary Fig. 2A–C). However, and somewhat unsurprisingly, there was a modest correlation of reperfusion rate with animal heart rate (Supplementary Fig. 2D).

Using intra-vital optical imaging measurements or microsphere deposition measurements, prior studies have shown increases and decreases in the islet blood-flow velocity in response to acute increases or decreases in the blood-glucose levels, respectively, with no changes occurring in the exocrine tissue[16,38,39]. Similar to these reports, we observed increases in the pancreas reperfusion rate (reflecting increased blood-flow velocity) associated with an acute increase in blood glucose following glucose delivery during an intraperitoneal (I.P.) glucose tolerance test (GTT, Fig. 1d, Supplementary Fig. 1J). We also observed significant decreases in pancreas reperfusion rate (reflecting

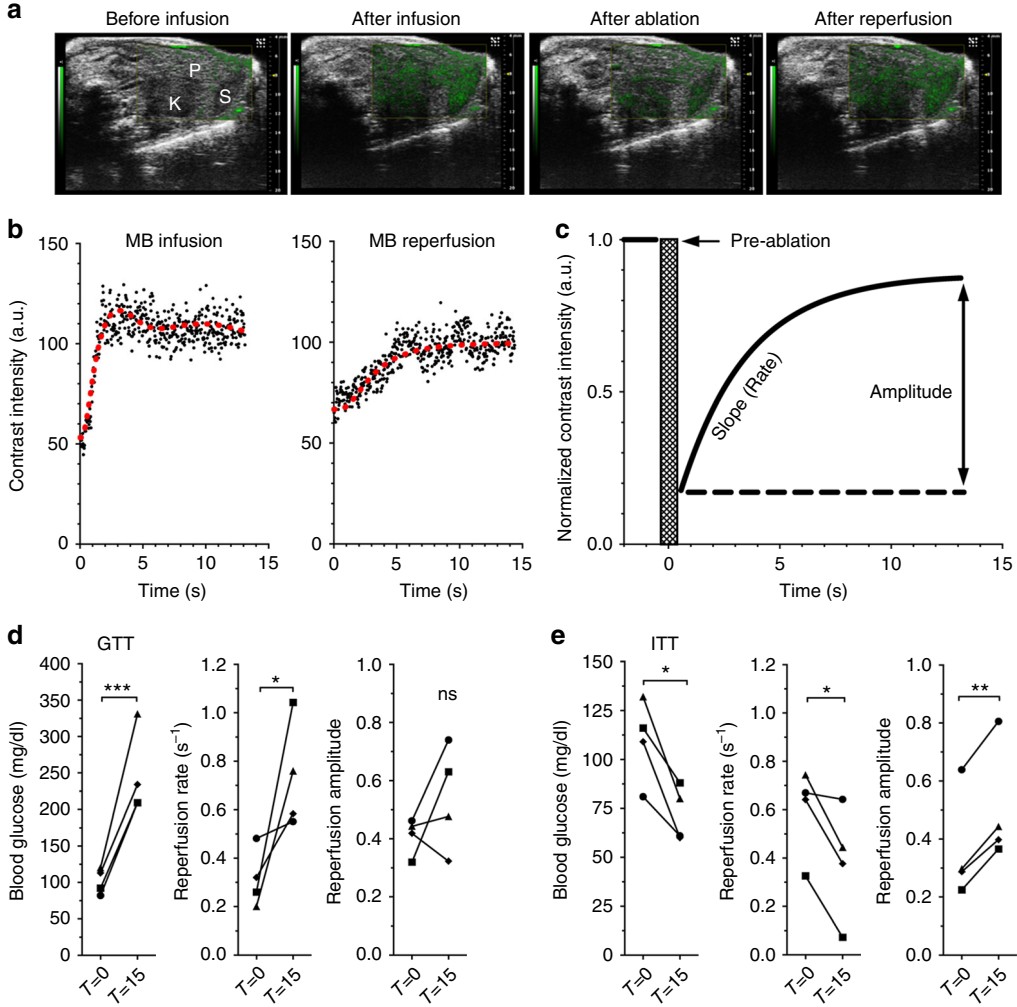

**Fig. 1** CEUS measurements and glucose-dependent changes in the pancreatic blood-flow dynamics in vivo. **a** Representative overlay images of NL contrast signal (green) on regular B-mode ultrasound image of pancreas (P), kidney (K), and spleen (S), before and after contrast infusion, and immediately following flash destruction (ablation) and reperfusion. **b** Representative time course of NL contrast signal infusion (left) and reperfusion following flash destruction (right). Red line indicates moving average of the signal. **c** Exponential rise fit of reperfusion data normalized to pre-flash destruction (pre-ablation) intensity. **d** Changes in blood glucose, pancreas reperfusion rate, and pancreas reperfusion amplitude before ($T = 0$) and 15 min after ($T = 15$) glucose delivery. **e** As in **d** before and 15 min after insulin delivery in C57BL/6 mice. $*p < 0.05$, $**p < 0.01$, $***p < 0.001$ comparing groups indicated (paired $t$-test data in **d**, **e**). Data in **d**, **e** represents $n = 4$ mice each, with each mouse indicated by a different symbol

decreased velocity) associated with an acute decrease in blood glucose following insulin delivery during an insulin tolerance test (ITT, Fig. 1e, Supplementary Fig. 1K).

These results establish the validity of CEUS to reliably measure pancreas blood flow dynamics. The consistency of the observed changes with established effects of acute changes to glucose levels on islet blood flow dynamics, but not exocrine blood flow, further supports these whole-pancreas CEUS measurements can reflect changes in the dynamics of islet blood flow.

**Changes in islet blood-flow dynamics upon STZ-induced islet injury**. To first establish whether CEUS can detect the islet blood-flow changes associated with the development of diabetes, we measured the pancreas blood reperfusion in female C57BL6 mice following multiple low-dose streptozotocin (STZ) treatment (5 daily doses 70 mg/kg). Multiple low-dose STZ treatment can induce β-cell injury leading to islet inflammation[40], and has previously been shown to alter the islet microvasculature[22]. STZ-treated animals showed robust increases in blood-glucose levels

2 weeks following the 70 mg/kg STZ treatment regime (Fig. 2a)[41]. CEUS measurements showed STZ-treated mice had significant increases in the reperfusion rate (Fig. 2b, c), but with little change in the reperfusion amplitude (Fig. 2d, Supplementary Fig. 3A).

To exclude any confounding effects of chronic hyperglycemia, which can itself cause microvasculature injury, we measured the pancreatic blood reperfusion kinetics in mice receiving a lower dose of STZ (5 daily doses 50 mg/kg) that is known to cause β-cell injury without overt hyperglycemia. After 2 weeks, these animals remained euglycemic, based on ad-lib glucose, but showed significant glucose intolerance (Fig. 2e) indicating β-cell damage and mimicking a pre-symptomatic ("pre-diabetic") disease state[42,43]. In these animals, we again observed a significant increase in reperfusion rate (Fig. 2f, g), with little change in reperfusion amplitude (Fig. 2h, Supplementary Fig. 2B). In these animals there was no significant correlation between the reperfusion rate and the degree of glucose intolerance (Supplementary Fig. 3C).

Therefore, the increased perfusion velocity in STZ treated animals, even in the absence of hyperglycemia (Supplementary

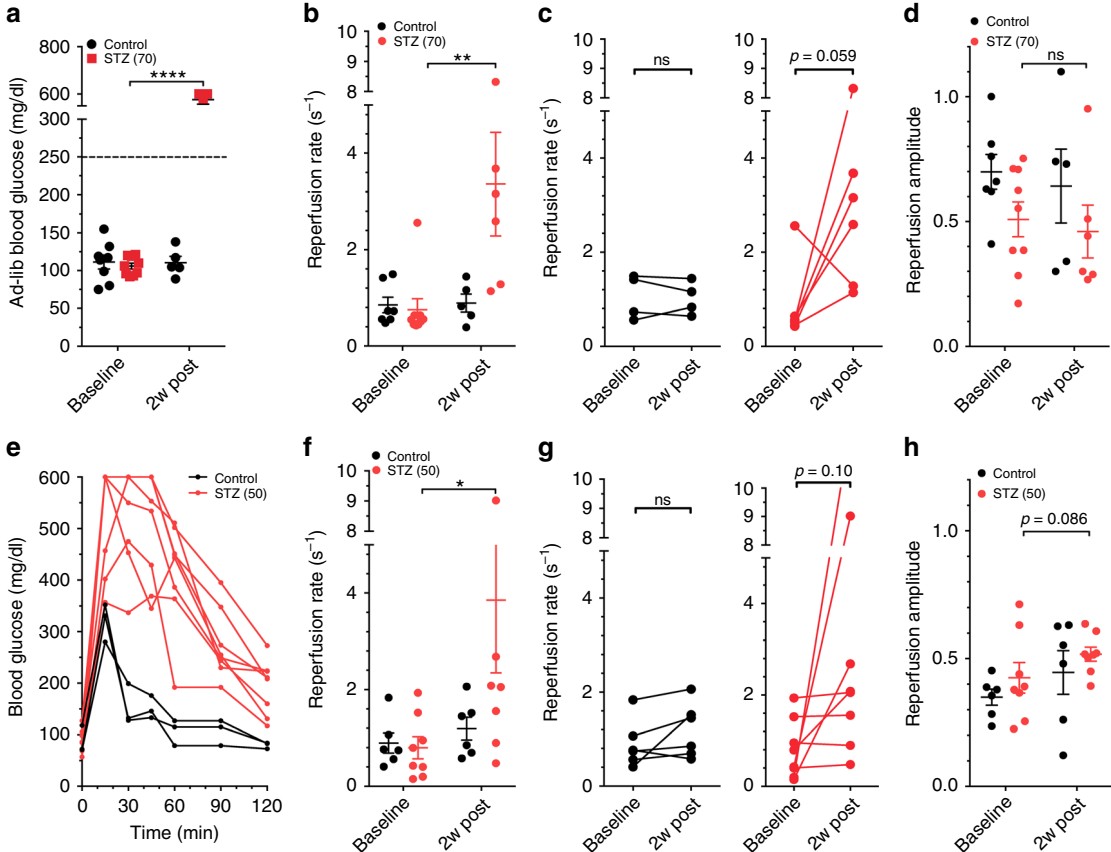

**Fig. 2** CEUS non invasively detects changes in the pancreas blood flow dynamics following STZ-induced β-cell injury. **a** Ad-libidum blood glucose concentrations of female C57BL/6 mice injected with 70 mg/kg STZ or buffer (control) before (baseline) and 2 weeks post treatment (2w post). **b** Reperfusion rate measured in mice in **a** treated with 70 mg/kg STZ or control before and 2 weeks post treatment. **c** Data in **b** showing changes in individual mice for reperfusion rate. **d** As in **b** measuring reperfusion amplitude. **e** Glucose tolerance tests from female C57BL/6 mice 2 weeks following treatment with buffer (control) or 50 mg/kg STZ. **f** Reperfusion rate measured in mice treated as in **e** with 50 mg/kg STZ or control before and 2 weeks post treatment. **g** Data in **f** showing the changes in individual mice for reperfusion rate. **h** As in **f** measuring reperfusion amplitude. Error bars represent s.e.m. * $p < 0.05$ comparing groups indicated (paired $t$-test for data in **c**, **g**; ANOVA for data in **a**, **b**, **f**, **h**). Data in **a–d** represents $n = 6$ STZ-treated ($n = 9$ at baseline for **b**, **d**) and $n = 5$ control mice ($n = 7$ at baseline for **b**, **d**); data in **e** represents $n = 8$ STZ-treated and $n = 3$ control; data in **f–h** represents $n = 8$ STZ-treated and $n = 6$ control mice

Fig. 4A,B), indicates that CEUS can detect changes in islet blood flow dynamics in vivo associated with β-cell injury.

**Changes in islet blood flow dynamics in NOD mice prior to diabetes onset**. STZ-induced hyperglycemia is a supra-physiological disease state, so we next determined whether CEUS could identify the pre-symptomatic stage of T1D using a more-relevant disease model. The non-obese diabetic (NOD) mouse develops spontaneous autoimmune diabetes as a result of infiltration of autoreactive T-cells into the islets of Langerhans, occuring many weeks before diabetes onset[44,45]. In our hands, female NOD animals consistently progressed to hyperglycemia between 13 and 35 weeks of age (Fig. 3a, Supplementary Fig. 5A). We measured pancreas blood reperfusion in female NOD mice at 6, 12, and 18 weeks of age, to respectively cover the stages early in disease, late in pre-symptomatic disease, and at a point where some animals had progressed to overt diabetes. At 12 weeks (pre-symptomatic diabetes) all NOD mice were euglycemic with the same ad-lib glucose level as at 6 weeks (early in disease), indicating an absence of confounding hyperglycemia (Fig. 3b). At 12 weeks, the reperfusion rates were significantly and substantially increased (Fig. 3c, d), while the reperfusion amplitudes were significantly decreased, compared to their 6 week scans

(Fig. 3e, f). At 12 weeks of age, mice on average showed a pattern of peri-insulitis to <50% insulitis, which is consistent with the increased reperfusion rates and decreased amplitude (Supplementary Fig. 6). We observed no additional change in islet blood reperfusion parameters from 12 to 18 weeks (Fig. 3c–f), including those animals that recently developed diabetes. This supports the idea that islet microvascular changes within NOD islets arises within the pre-symptomatic disease stage, and is not associated with disease onset and hyperglycemia (Supplementary Fig. 4C, D)[22,46]. To confirm that changes in pancreas reperfusion parameters in NOD animals were disease dependent, we examined immuno-deficient NOD-RAG1 knockout (NOD-RAG1$^{-/-}$) mice that do not develop insulitis or diabetes. These animals showed no change in blood reperfusion rate or amplitude over the corresponding time points (Fig. 3g–i, Supplementary Fig. 5B). We also observed a lower infusion steady state NL signal in NOD mice during disease progression, suggesting a reduction in islet vasculature, with no significant change in NOD-RAG1$^{-/-}$ mice (Supplementary Fig. 5C). The flux of blood flow (rate × amplitude) showed a modest elevation in NOD mice during disease progression (Supplementary Fig. 5D).

These results highlight that changes in islet blood flow in NOD mice are associated with the progression of diabetes, suggesting that increases in islet blood flow could be a predictive marker of

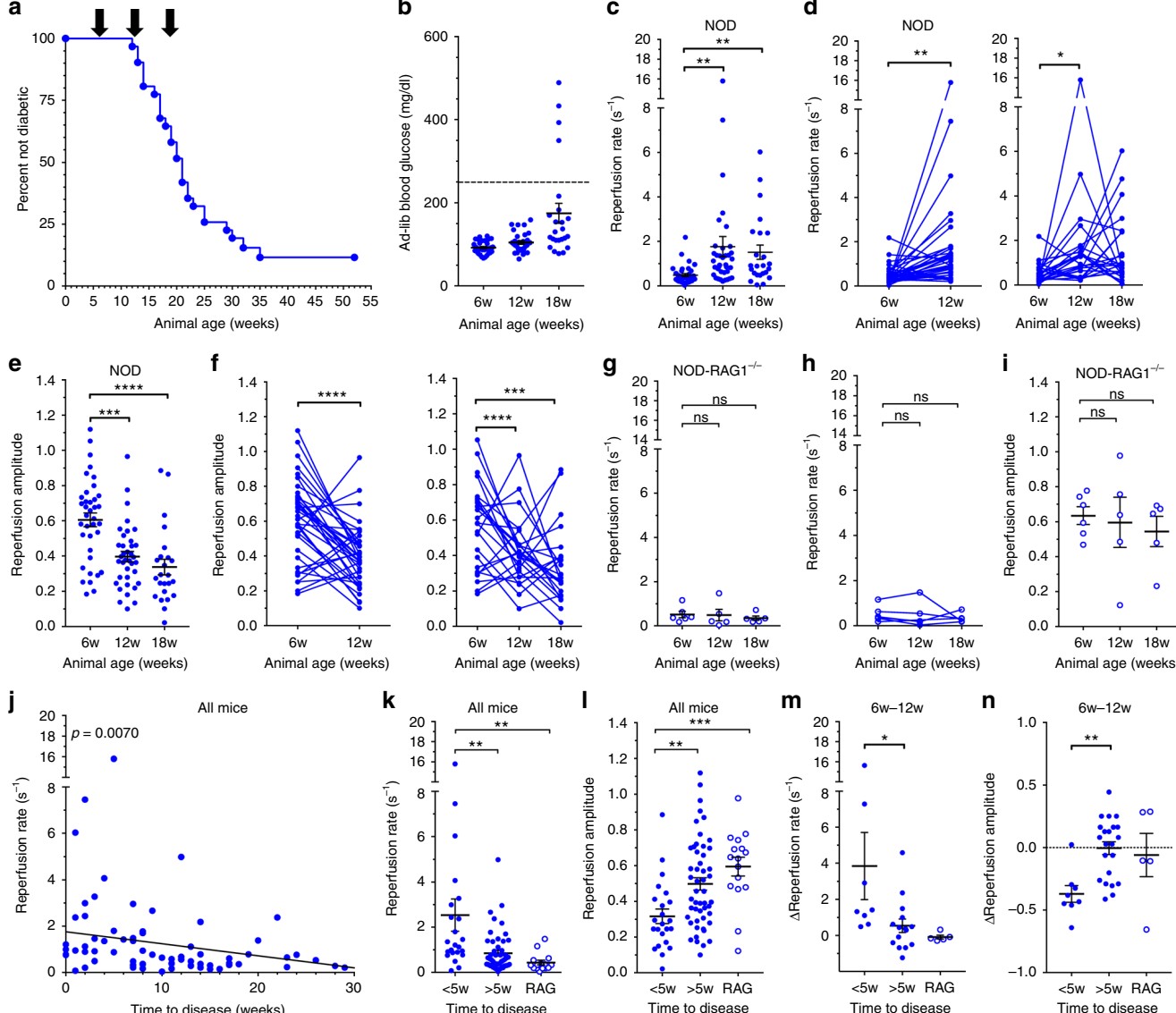

**Fig. 3** CEUS non-invasively detects changes in pancreas blood flow dynamics associated with the progression of diabetes in NOD mice. **a** Survival curves indicating time of diabetes onset for all NOD animals analyzed for this study. **b** Ad lib blood-glucose levels at the time of measurement for all NOD mice. **c** Reperfusion rate measured in NOD mice at ages indicated. **d** Data in **b** showing changes in individual NOD reperfusion rates between 6 and 12 weeks of age (left), or 6 to 18 weeks (right). **e** As in **c** for reperfusion amplitude. **f** As in **d** for changes in the reperfusion amplitude between 6 and 12 weeks of age (left), or 6 to 18 weeks (right). **g** Reperfusion rate measured in NOD-RAG1 KO immunodeficient mice at ages indicated. **h** Data in **g** showing changes in individual NOD-RAG1 KO reperfusion rates between 6 and 18 weeks. **i** As in **g** for reperfusion amplitude. **j** Correlations of reperfusion rate with time to diabetes from CEUS scan, in weeks. **k** Average reperfusion rate over animals that progressed to disease <5 weeks or >5 weeks from CEUS scan, along with measurements in NOD-RAG1 KO mice for comparison. **l** As in **k**, for average reperfusion amplitude. **m** Average change in the reperfusion rate between 6 and 12 weeks averaged over animals that progressed to disease <5 weeks or >5 weeks from CEUS scan, along with average changes in NOD-RAG1 KO mice for comparison. **n** As in **m** for change in reperfusion amplitude between 6 and 12 weeks. Error bars represent s.e.m. *$p < 0.05$, **$p < 0.01$, ***$p < 0.001$, ****$p < 0.0001$ comparing groups indicated (paired $t$-test data in **d**, **f**, **h**; unpaired $t$-test in **k–n**; ANOVA data in **c**, **e**, **g**, **i**). Data in **a–f** represents $n = 37$ NOD mice ($n = 24$ for 18 weeks), data in **g–i** represents $n = 6$ NOD-RAG1 KO mice ($n = 5$ at 12w, 18w), data in **j–l** represents 71 scans over 27 NOD mice, data in **m**, **n** represents $n = 27$ NOD mice. A mixed-effects model was used to assess the statistical significance and generate the regression in **j**

disease onset. We assessed correlative trends between the time to diabetes onset from a CEUS scan and the reperfusion rate or amplitude from that scan. Population-wide analysis showed that increased pancreas reperfusion rates (Fig. 3j) and decreased pancreas reperfusion amplitudes (Supplementary Fig. 5E) were significantly correlated with a faster progression to diabetes. Animals that progressed to diabetes within 5 weeks of the CEUS scan showed a significantly larger rate (~4.5 fold increase) and lower amplitude (~20% decrease), compared to animals that

progressed to diabetes >5 weeks (Fig. 3k, l). Furthermore, a larger increase in the reperfusion rate or larger decrease in the reperfusion amplitude between 6 and 12 weeks correlated with a faster progression to diabetes (Supplementary Fig. 5F, G), where animals that progressed to diabetes within 5 weeks of the 12 week CEUS scan showed a significantly greater change in the reperfusion rate and the amplitude between 6 and 12 weeks, compared to animals that progressed to diabetes >5 weeks (Fig. 3m, n). Changes in reperfusion rate and amplitude between

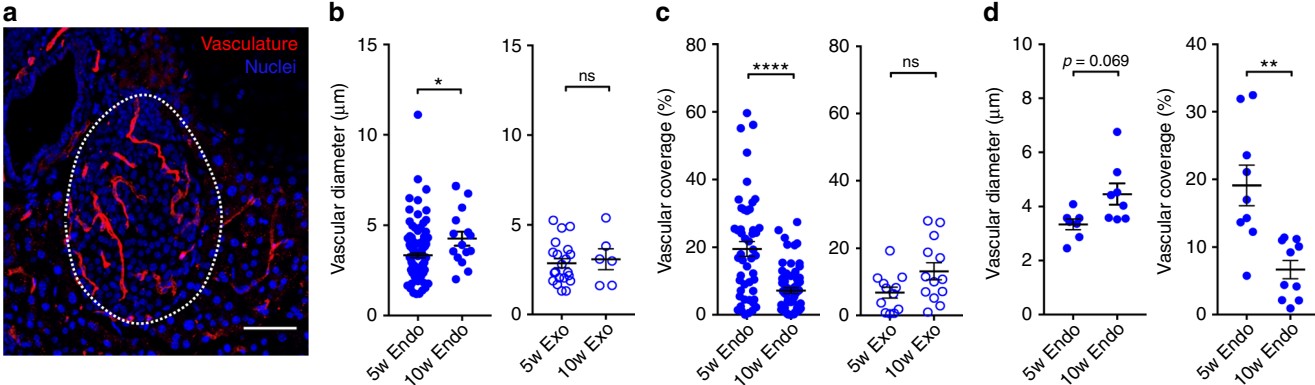

**Fig. 4** Changes in islet, but not exocrine microvascular morphology, with age in the pancreata of NOD mice. **a** Representative maximum-projection confocal image over 10 μm depth of pancreas section from NOD mouse infused with texas-red labeled tomato lectin (red). Islet is circled with a dotted line, as determined from brightfield and DAPI-labeling morphology. **b** Mean vessel diameter in islet (Endo) and exocrine tissue (Exo) in 5- and 10-week-old NOD female mice. **c** As in **b** for vascular coverage. **d** Data for islet vessel diameter and vascular coverage in **b**, **c** plotted by mouse. Scale bar in **a** represents 100 μm. Error bars represent s.e.m. *$p < 0.05$, **$p < 0.01$, ***$p < 0.001$, ****$p < 0.0001$ comparing groups indicated (t-test data in **b–d**). Data in **b** represents in total 65 islets (37 islets, at 5w, 28 islets at 10w) and 27 exocrine regions (21 at 5w, 6 at 10w) from 7 NOD mice per time point. Data in **c** represents in total 112 islets (49 at 5w, 63 at 10w) and 25 exocrine regions (12 at 5w, 13 at 10w) from 9 NOD mice per time point

12 and 18 weeks did not show a significant correlation with faster progression to diabetes (Supplementary Fig. 5H–K). We tested for any predictive ability by performing ROC analysis (Methods) to optimally separate measurements between NOD and NOD-RAG1$^{-/-}$ mice. Using this separation as a prediction of the disease level, mice that were predicted to be disease positive, based on their reperfusion rate showing a large increase, or amplitude showing a large decrease from 6 to 12 weeks, progressed to diabetes significantly more rapidly than mice predicted to be disease negative (Supplementary Fig. 7A–C).

To further exclude any confounding effects of hyperglycemia, we measured pancreas blood reperfusion in db/db mouse model of T2D, which develops profound insulin resistance and chronic hyperglycemia. In db/db animals, blood glucose was mildly elevated at 4 weeks of age and dramatically elevated at 8 and 12 weeks (Supplementary Fig. 8A, 4E). At 8 weeks, the reperfusion rates in db/db mice showed a small, but nonsignificant trend to elevation, compared to at 4 weeks. However at 12 weeks, the reperfusion rates were similar to those values observed at 4 weeks, as well as those observed in control animals (Supplementary Fig. 8B). At all measurement times, the amplitude was unchanged (Supplementary Fig. 8C).

Therefore, increased perfusion velocity (increased reperfusion rate) and reduced perfusion volume (reduced reperfusion amplitude) occurs in NOD animals undergoing moderate levels of insulitis prior to diabetes onset and independent of hyperglycemia. These measures of pancreatic blood flow redistribution were well separated from controls and predictive of disease onset, suggesting diagnostic potential.

**Changes in islet microvascular morphology in NOD mice**. We next tested whether the observed increases in islet reperfusion rate and decreases in reperfusion amplitude in NOD animals were consistent with the morphological changes in islet microvasculature associated with disease progression. We performed histological analysis of the islet and the exocrine microvasculature from NOD mice at 5 and 10 weeks of age (the slightly reduced time points reflecting the slightly more rapid progression to diabetes in the NOD colony used for these measurements) (Fig. 4a). In cryopreserved tissue sections, we identified islets and exocrine tissue distant from islets, and analyzed the microvascular morphology. Within the islets, we observed a significant increase in vessel diameter at 10 weeks of age compared to 5 weeks of age,

whereas no significant change occurred in the exocrine tissue (Fig. 4b). In contrast, we observed a reduced area of vasculature coverage within the islets at 10 weeks compared to 5 weeks, with no significant change in the exocrine tissue (Fig. 4c). These trends held true when examining the morphology by mouse (Fig. 4d). Thus, the changes in islet microvasculature morphology observed prior to diabetes onset are consistent with the CEUS reperfusion measurements: There exist fewer vessels that have greater diameter, leading to a reduced perfusion volume, but greater flow velocity. This is also consistent with the lower vascular coverage within islets previously shown in recently diabetic NOD mice[24,25,47].

**Changes in islet blood-flow dynamics predicts efficacy of immunotherapy**. While preventative therapies for T1D hold clinical promise if initiated early in T1D progression[48], there are limited means to monitor the treatment efficacy. Therefore, we next tested whether CEUS could detect changes in disease progression during pharmacological intervention, by utilizing an adoptive transfer (AT) model of inducible diabetes. This model shows well-defined T-cell-mediated autoimmune diabetes that is more consistent and rapid than in NOD mice, allowing convenient assessment of therapeutic reversal[49]. Splenocytes (~20 million) isolated from recently diabetic female NOD donors were injected into female NOD-Scid recipients ("AT"). All AT animals receiving diabetogenic splenocytes progressed to diabetes within 3–6 weeks (Fig. 5a, Supplementary Fig. 9C). CEUS measurements of pancreas blood reperfusion showed higher well-separated reperfusion rates in AT mice receiving splenocytes 2 weeks and 4 weeks following transfer compared to vehicle control animals (Fig. 5b, c). However, we observed no significant change in reperfusion amplitude in AT mice (Fig. 5d, Supplementary Fig. 9D). At each time point, the measured mice remained euglycemic (Supplementary Fig. 4F).

We next tested whether the efficacy of pharmacological intervention could be monitored by changes in islet blood-flow dynamics. At 2 weeks post delivery of splenocytes, AT animals received a single dose (20 mg) of antiCD4 antibody[50] to deplete CD4$^+$ T-cells (Supplementary Fig. 9A, B), which delayed and blunted diabetes onset (Fig. 5e, Supplementary Fig. 9C). Importantly, and enabling us to test whether CEUS can detect any responses to therapeutic intervention, a subset of animals responded to antiCD4 treatment and maintained euglycemia

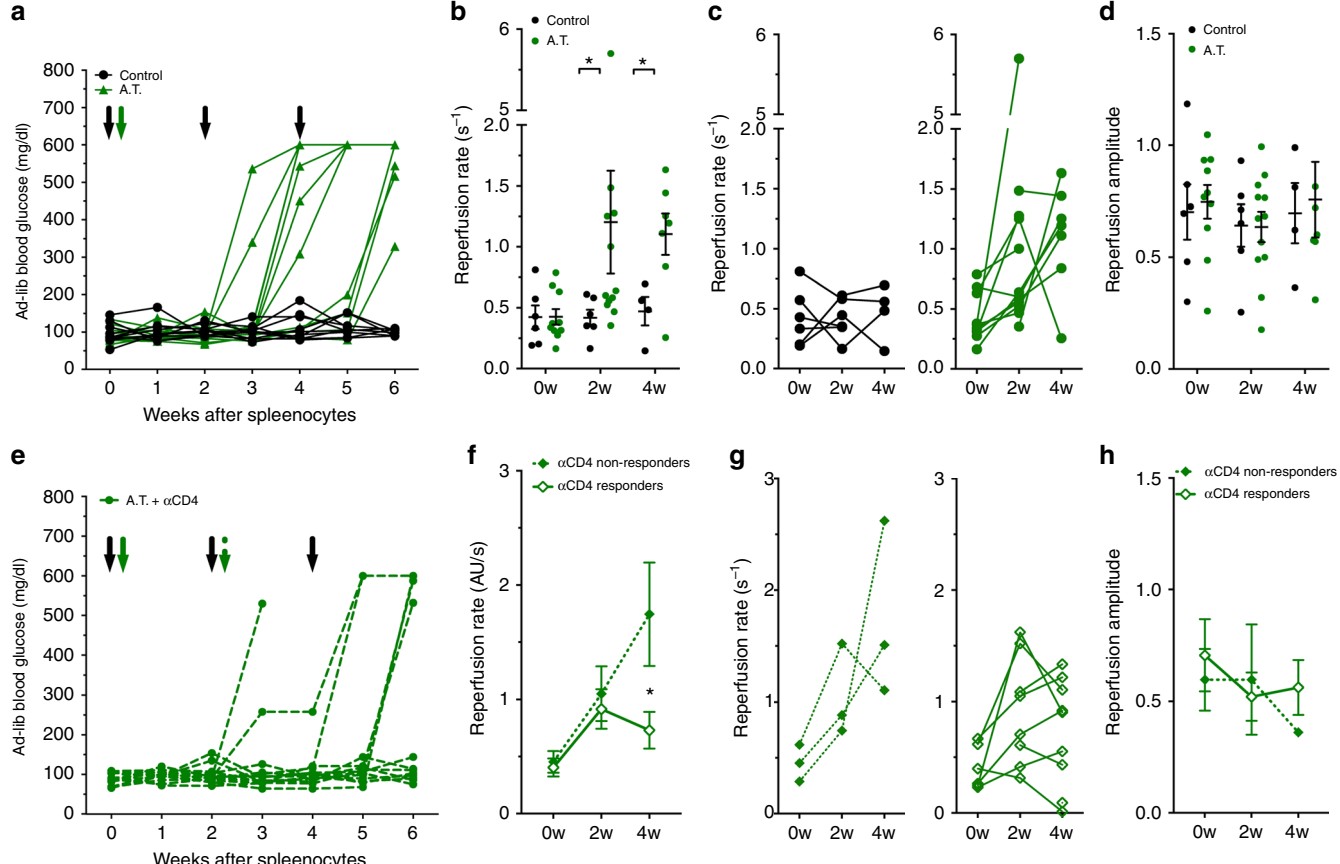

**Fig. 5** CEUS non-invasively detects changes in pancreas blood flow dynamics associated with immunomodulatory halting of disease. **a** Ad-lib blood-glucose time courses in adoptive transfer animals treated with splenocytes from diabetic female NOD donors (A.T., green) or treated with buffer alone (Control, black). Black arrows indicate time points of CEUS scans, green arrow indicates delivery of splenocytes. **b** Reperfusion rate in A.T. animals (green) or control animals (black) at baseline (0w), 2 weeks (2w) and 4 weeks (4w) post splenocyte or vehicle transfer. **c** Data in **b** showing changes in individual mice for reperfusion rate. **d** As in **b** for reperfusion amplitude. **e** Ad-lib blood-glucose time courses of A.T. animals that were treated with 20 mg anti-CD4 antibody. Black arrows indicate time of CEUS scans, green solid arrow indicates time of splenocyte delivery, green striped arrow indicates time of anti-CD4 treatment. Animals that progressed to hyperglycemia within 6 weeks were denoted as non-responders. **f** Average reperfusion rate in anti-CD4 responders (open diamonds) and non-responders (closed diamonds) before (0w), 2 and 4 weeks post splenocyte transfer. **g** Data in **f** showing changes in individual responder and non-responder mice for reperfusion rate. **h** As in **f** for reperfusion amplitude. Error bars represent s.e.m. *$p < 0.05$, **$p < 0.01$, ***$p < 0.001$ comparing groups indicated (paired *t*-test for data in **c**, **g**; unpaired t-test for data in **f**, **h**; ANOVA for data in **b**, **d**. Data in **a–d** represents $n = 11$ AT mice and $n = 6$ control mice, data in **e–h** represents $n = 9$ responder mice and 3 non-responder mice

for >6 weeks post splenocyte delivery ("responders"), while the remaining animals progressed to diabetes ("non-responders"). At both 2 weeks and 4 weeks, all responders and non-responders were euglycemic (Supplementary Fig. 4F). AntiCD4 responders showed a significantly lower reperfusion rate compared to antiCD4 non-responders, with a blunting of the increasing reperfusion rate associated with disease progression (Fig. 5f, g). However, non-responders showed a progressive elevation in reperfusion rate that was similar to that in untreated AT mice. The reperfusion amplitude also showed a trend of decline in non-responders and elevation in responders (Fig. 5h, Supplementary Fig. 9E).

Across all antiCD4-treated animals, only 1/3 non-responder but 8/8 responders showed a change in reperfusion rate between 2 and 4 weeks that was less than the lowest quartile change observed in untreated AT mice., thus This shows strong separation between responders and non-responders. Again, following ROC analysis, antiCD4 treated mice that were predicted to be disease positive based on their reperfusion rate showing a large increase from 2 to 4 weeks, progressed to diabetes more rapidly and with greater incidence than mice predicted to be disease negative (Supplementary Fig. 7D–F).

Therefore, the increased perfusion velocity in the AT model can successfully detect disease progression and predict disease reversal upon therapeutic treatment.

## Discussion

There are limited approaches for reliable and reproducible diagnosis of T1D during the pre-symptomatic phase of disease progression given i) the lack of clinical presentation of symptoms until significant β-cell death has occurred, ii) heterogeneity in underlying disease progression, and iii) a technical inability to monitor β-cell decline and insulitis over time. The approach we describe here is the first, to our knowledge, to establish the ability to use ultrasound imaging modalities to non-invasively detect and track the progression of pre-symptomatic T1D and its therapeutic reversal in pre-clinical models. We demonstrated that islet blood perfusion velocity substantially increases, while perfusion volume generally decreases prior to diabetes onset. Importantly these changes were well-separated from controls, and predicted both rapid onset of diabetes in NOD mice and successful (or unsuccessful) immunomodulatory intervention to halt diabetes in AT animals (Figs. 3, 5; Supplementary Fig. 7). The qualitatively

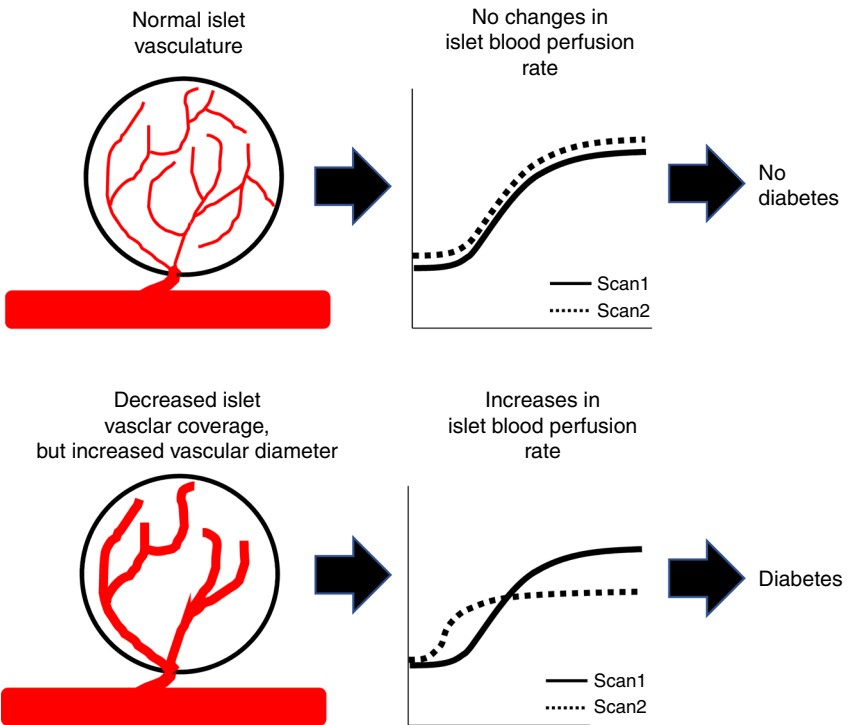

**Fig. 6** Summary by which CEUS measurement of the pancreatic blood-flow dynamics can be used to predict diabetes onset and immunotherapy efficacy. (Top) In an animal that is not progressing to diabetes, there is no change in the reperfusion rate or the amplitude over time, consistent with normal islet microvasculature organization. (Bottom) In an animal that is in pre-diabetes and/or progressing to overt disease, consecutive CEUS scans will show an increase in reperfusion rate, and potentially a decrease in reperfusion amplitude. This change is associated with a decrease in the islet vascular coverage, yet an increase in diameter of the remaining vessels. Thus we suggest that changes in the reperfusion rate (increases) indicate likely progression to diabetes. Furthermore, a defined threshold may also identify an "abnormal" reperfusion measurement to indicate the likely progression to diabetes, dispensing with "scan 1"

similar results achieved in all T1D models, together with advanced prediction of disease progression or disease reversal, highlights the disease specificity, and thus the power of CEUS islet blood-flow measurements. Our measurements were also sensitive enough to monitor disease progression, as demonstrated by detection of significant islet blood-flow changes associated with (1) mild STZ-induced β-cell injury that does not cause hyperglycemia (Fig. 2); (2) NOD mice, where peri-insulitis is predominant (Supplementary Fig. 6); and (3) in NOD and AT models many weeks before diabetes onset (Figs. 3, 5). However, measurements in NOD mice at 5–6 weeks, where mild insulitis likely occurs[51], was not different from immune-deficient controls, indicating that some minimum limit of detection does exist. Taken together, our CEUS measurements of pancreas blood-flow dynamics can non-invasively and longitudinally track the underlying disease in pre-symptomatic T1D in the pre-clinical models employed. This suggests that CEUS could be used as a clinical diagnostic to identify both diabetes progression and diabetes reversal upon therapeutic intervention (Fig. 6). A key future goal will be to perform prospective pre-clinical studies to test whether CEUS measurements of pancreas blood flow dynamics can predict the efficacy of therapeutic agents currently under trial to prevent T1D.

The CEUS approach we describe compares favorably with existing imaging approaches reported in pre-clinical T1D models. While in vivo β-cell labeling techniques have shown promise in detecting the changes in β-cell mass in STZ or NOD animal models[52–54], a clear challenge is the exquisite specificity needed for the small β-cell volume to overcome any exocrine-originating background signal. An advantage of our approach is that the well-perfused islet microvasculature is heavily enriched compared to the poorly perfused exocrine vasculature, and that the islet

perfuses on a rapid time scale[38], greater than that in the exocrine compartment[20]. Further, the CEUS measurements of reperfusion rate and amplitude both showed changes associated with diabetes progression, with the reperfusion rate showing robust changes across all models examined. Measuring two independent parameters, as performed here, may also provide greater specificity and prediction for disease development compared to measuring a single signal enhancement/diminishment. Demonstrating the changes in diabetes models, but also separating in advance mice that progressed or did not progress to diabetes is also a powerful validation of the CEUS approach we describe, which is often lacking with other published approaches. Islet microvascular permeability, as measured by magnetic nanoparticle incorporation, has been reported to detect progression of diabetes and reversal upon immunotherapy[12,55], with additional demonstration in patients[11,56]. However, the CEUS method outlined here may provide a more convenient approach for clinical deployment (see below).

We did observe a spread in measurements in all animal models used in this study. In the AT model where disease progression is more uniform, we observed clear separation between AT mice prior to disease onset and controls (Fig. 5b). In NOD animals, the spread observed at later time points likely results from heterogeneity in disease progression, as supported by the association between reperfusion rate or amplitude and their changes with the time to diabetes onset (Fig. 3). Further, in NOD mice we observed diverse temporal patterns (Fig. 3d) including more frequent scan time points that may reveal the complex patterns of disease progression and resolve disease heterogeneity. For example, in humans, "waves" of β-cell destruction have been suggested, rather than a gradual decline[57]. Nevertheless, we anticipate that even a small number of longitudinal "snap shots"

(Fig. 3m, n) may be useful for monitoring the effectiveness of potential therapeutic treatments in disease prevention studies, where therapeutic intervention occurs during the pre-symptomatic phase of T1D.

The non-invasive CEUS measurements we made detected changes in islet blood-flow dynamics that were consistent with the islet microvascular remodeling we observed (Figs. 3, 4). An increased reperfusion rate (indicating increased velocity) is consistent with the increases in vessel diameter, while the reduced perfusion amplitude (indicating reduced volume) is consistent with the reduction in vascular coverage. Invasive microsphere deposition or optical imaging studies have shown qualitatively similar blood flow changes in the islets in animal models of diabetes[18,21,23]. Changes in the islet microvasculature, but not exocrine vasculature, have also been observed in animal models of T2D[26,58] and in human donors with T2D[27]. Islet inflammation is common to all of the animal models we examined, and inflammation is well established to lead to altered tissue blood flow. Notably, a recent study using CEUS also demonstrated differences in perfusion in a BB rat model of T1D, likely as a result of inflammation[59]. In models of T2D, it has been suggested that altered islet microvascular diameter results from altered islet blood flow rather than being a cause. However, the precise link between inflammation, altered microvasculature organization, and islet blood flow in the models of T1D examined here remains to be determined.

We also inferred that measurements of pancreas reperfusion dynamics reflect islet blood-flow dynamics, based on previous measurements showing that islet perfusion and pancreas perfusion highly correlate[20]. This is supported by our observations that blood-flow dynamics change upon acute increases or decreases in glucose, which has previously been shown to specifically affect islet blood flow with no effect on exocrine blood flow[16,38,39]. However, it is important to note that NOD and AT models of T1D were euglycemic at the time points examined prior to diabetes onset, and therefore glycemic state cannot explain the changes in blood flow dynamics we observed (Supplementary Fig. 3)[51]. Each model we studied also involves islet-specific injury or infiltration, and we did not observe significant changes in the exocrine vascular morphology (Fig. 4). However, we note that some changes in the exocrine tissue have been observed in human T1D[60,61], although little has been reported in pre-symptomatic T1D[62]. Changes in pancreas volume would not be expected to significantly impact the more predictive reperfusion rate, but may affect the reperfusion amplitude. Irrespective of the precise mechanisms involved, we have demonstrated that measuring changes in the blood-flow dynamics across the pancreas reflect changes in the islet over the exocrine tissue, and assess T1D progression in the models examined.

Given the lack of validated clinical approaches to follow pre-symptomatic T1D, a key advantage of the CEUS approach is the ability to readily translate to the clinic. Ultrasound systems are readily deployable, convenient to use, and relatively inexpensive. Importantly, ultrasound contrast agents are FDA approved for echocardiography measurements (DEFINITY, Lantheus) and liver measurements including in pediatric populations (LUMA-SON, Bracco). Off-label uses have been reported in the pancreas for nondiabetes applications[33–35]. While we utilized a small animal imaging machine employing higher frequencies than used clinically, similar measurements have been performed in humans at the lower frequencies used clinically and within mechanical index safety limits, including pancreas imaging[33–35]. We acknowledge that all animal models display some limitations in faithfully modeling human T1D, and differences exist between mouse and human microvasculature that warrant consideration. Data examining whether the islet microvasculature changes during human T1D progression prior to onset is lacking: In established T1D following loss of β-cell mass, there is an increase in islet microvascular density, but a decrease in vessel diameter with no change in exocrine vasculature (M. Campbell-Thompson, personal communication), suggesting that the pattern of blood-flow changes with T1D onset may differ between NOD mice and humans. However, the islet microvasculature changes in T2D are similar between mouse and human[26,27], and microvascular permeability is similarly altered in both mouse and human recent onset T1D[56]. It is also unknown, how total pancreas blood flow relates to changes in the islet microvasculature and the blood flow in human, especially given the differences between mouse and human islet microvascular coverage, organization, and integration with surrounding exocrine tissue[63–65]. In addition, given human variability, islet blood-flow dynamics may vary between individuals due to age, insulin resistance, cardiac function, or other factors. Such subject-to-subject variability poses challenges for any approach translated to human studies, and has likely limited clinical translation to date. This indicates a need for longitudinal measurements, which is a requirement ideally suited for ultrasound modalities.

In conclusion, we present a readily deployable, non-invasive, longitudinal imaging method for the diagnosis and longitudinal monitoring of islet blood-flow changes associated with diabetes progression in pre-clinical models of T1D. We show that changes in islet microvasculature and islet blood flow can be exploited to predict the early onset of diabetes and to monitor the efficacy of immunomodulatory therapies in the treatment of T1D. With this data, we present a rationale for testing CEUS measurements of islet blood flow as a clinically deployed diagnostic imaging technique. This may address the major shortage of approaches to track the disease progression in human patients with pre-symptomatic T1D, and those receiving preventative treatments.

## Methods

**Animals.** All animal procedures were performed with ethical approval and in accordance with guidelines established by the Institutional Animal Care and Use Committee of the University of Colorado. 12-week-old female C57Bl/6 mice were purchased from Envigo (formerly Harlan). Female NOD mice were either purchased from Jackson Laboratories (Bar Harbor, ME) or bred in-house. Female NOD-Rag1$^{-/-}$ mice were bred in-house. NOD-Scid animals were purchased from Jackson Laboratories. Throughout the study, animals were monitored weekly for blood glucose concentration utilizing a glucometer (Bayer).

**CEUS imaging.** General anesthesia was established with isoflurane inhalation for a total of 15–20 min for all animal strains. Prior to CEUS imaging, a custom-made 27 G 1/2″ winged infusion set (Terumo BCT, Lakewood, CO) attached to polyethylene tubing (0.61 OD × 0.28 ID; PE-10, Warner Instruments) was inserted in the lateral tail vein and secured with VetBond (3M). The abdominal fur was removed using depilatory cream, and ultrasound coupling gel was placed between the skin and transducer. Foot pad electrodes on the ultrasound machine platform monitored the animal's electrocardiogram and breathing rate. All animals were constantly monitored throughout the imaging session to maintain body temperature (36.4 ± 0.7 °C, $n = 88$) and respiration rate (89.8 ± 2.5 min$^{-1}$, $n = 88$).

A VEVO 2100 small animal high-frequency ultrasound machine (Visual Sonics, Fujifilm, Toronto, Canada) was used for all experiments. For CEUS imaging, a MS250 linear array transducer was used at a frequency of 18 MHz. Normal B-mode imaging (transmit power 100%) was performed prior to the contrast infusion to positively identify anatomy of the pancreas body and tail, based on striated texture and location in relation to spleen, kidney, and stomach (Fig. 1a). NL contrast mode was initiated following positive identification of the pancreas and selection of a region of interest. For NL imaging, acquisition settings were set to the following: transmit power 10% (MI = 0.12), frequency 18 MHz, standard beamwidth, contrast gain of 30 dB, and 2D gain 18 dB, with an acquisition rate of 26 frames per second. "Gating", to account for breathing artifacts, was performed manually.

Size-isolated microbubble contrast agent ("SIMB3-4", Advanced Microbubble Laboratories, Boulder, CO) was injected as a single bolus of ~10 million bubbles in phosphate buffered saline (pH 7.4) in the lateral tail vein via the catheter. SIMBs were allowed to circulate throughout the animal for ~20 s to reach a relative steady state of systemic distribution. The SIMB destruction was initiated by delivery of a high mechanical index pulse (VEVO2100 burst mode, MI = 0.2), to destroy a portion of SIMBs within the imaging plane[66,67]. Data were acquired for at least 10 s

following SIMB destruction to adequately measure reperfusion into the tissue. While 1–2 μm SIMBs may fall closer to resonance on the VEVO2100 small animal machine, the cross section of 3–4 μm SIMBs will be substantially larger, given the dependence of cross section on $r^6$.

For analysis of NL contrast intensity reperfusion kinetics, the background NL intensity taken before SIMB infusion was subtracted from the entire trace. Each reperfusion time-course was normalized to a 0.5 s average of the steady state NL contrast intensity, immediately prior to flash destruction. The resultant normalized reperfusion curves were fit with an exponential rise equation, $F(t) = C + A(1-e^{-kt})$, where $C$ is the offset from zero, $A$ is the amplitude of the curve, and $k$ is the rate of reperfusion (Fig. 1c). CEUS measurements were excluded from analysis if a poor infusion was recorded and/or poor microbubble flash-destruction occurred (and thus poorly defined recovery).

**Streptozotocin treatment.** Adult female C57Bl/6 mice (20–25 g, Harlan at 12 weeks of age) received 5 × daily low-dose streptozotoxin (STZ) treatment, as previously described[29]. Animals were fasted for 6 h before receiving an I.P. injection of 70 mg/kg or 50 mg/kg STZ (Sigma Aldrich, St. Louis, MO) dissolved in fresh 0.1 M sodium citrate buffer, pH 4.5. Age-matched control animals were injected with citrate buffer alone and treated in the same manner as STZ-injected animals.

**Isolation and adoptive transfer of diabetogenic splenocytes.** Splenocytes were isolated from diabetic female NOD mice (hyperglycemic for < 1 week), manually dissociated and counted in ice-cold HBSS (without MgCl₂ and CaCl₂). Leukocytes were counted for an estimate of cellular density. 12–14-week-old NOD-Scid immunodeficient recipients received a single I.P. dose of $20 \times 10^6$ leukocytes resuspended in HBSS. Control animals were injected with equivalent volumes of HBSS without leukocytes. A subset of NOD-Scid animals that had undergone splenocyte adoptive transfer were injected with a single dose of 20 mg anti-mouse CD4, clone GK1.5 (BP0003-1; BioXCell, W. Lebanon, NH) 2 weeks following splenocyte delivery.

**Anti-CD4 immunotherapy & flow cytometry.** CD4+ T-cell diminishment was confirmed via flow cytometry of CD4+ T-cells. Antibodies used for flow cytometry: FITC-CD8 53.6 (BD Pharmingen #553031); APC-CD4 RM4-5 (BioLegend #100516), PE-CD9. All cells analyzed in flow were confirmed as immune via PerCP Cy5.5 CD45.1 A20 (BD Pharmingen #560580). The gating strategy to isolate percent CD4+ T-cells proceeded as: (a) gate on forward scatter high, side scatter medium-high population; (b) gate on CD45.1 positive (PerCP-Cy5.5), CD4 positive population (APC); (c) CD8 (FITC) and CD19 (PE) were unused.

**Detailed description of CEUS imaging paradigms.** GTT and ITT (Fig. 1): Animals were baseline scanned before an I.P. injection of glucose (2 g/kg) or insulin (0.75 U/kg). Animals were maintained at proper temperature, respiration, and anesthesia for 15 min. Blood-glucose concentration was assessed prior to performing another CEUS scan. Only animals with a glucose deviation of at least 20% were analyzed (Fig. 1e, f). Each animal acted as its own internal control, and comparisons were made within the individual animals.

STZ-induced beta-cell damage (Fig. 2): Animals were baseline scanned ("Baseline") followed by multiple low-dose treatments of STZ (either 50 mg/kg, 70 mg/kg, or buffer alone, as noted). Two weeks following the STZ treatment week ("2w Post"), animals were CEUS scanned. Each animal acted as its own internal control, and comparisons are made within individual animals, as well as comparisons made between all experimental groups of animals.

NOD progression to disease (Figs. 3, 4): CEUS was performed in NOD (or NOD-RAG1⁻/⁻ controls) mice at 6, 12, and 18 weeks of age (for in-house bred animals, mean diabetes onset is 19 ± 2 weeks) or at 5, 10, 15 weeks of age (for mice purchased form Jackson laboratories, mean diabetes onset is 16 ± 2 weeks), as noted. All animals were monitored weekly for ad lib blood-glucose concentration, and were euthanized if presented with two consecutive glucose measurements greater than 250 mg/dl. In a subset of animals that were utilized for insulitis scoring (see below, Supplementary Fig. 4), animals received 6 and 12 week age scans followed by processing for histological analysis (see below). All NOD/NOD-RAG1⁻/⁻ animals in this study were followed for one year, or until the animal presented with two consecutive weekly glucose measurements greater than 250 mg/dl. Paired comparisons were made within individual animals, and comparisons were made between all experimental groups of animals.

DB/DB progression to disease (Supplementary Fig. 7): CEUS was performed in db/db (or C57BLKS controls) mice at 4, 8, and 12 weeks of age, as noted. For C57BLKS controls at 8 weeks, data is excluded as a result of faulty data collection. All animals were monitored weekly for ad lib blood-glucose concentration. Comparisons were made between all experimental groups of animals.

Adoptive transfer & anti-CD4 (Fig. 5): NOD-scid immune-deficient animals were baseline scanned ("baseline"), followed by I.P. injection of 20 million leukocytes from a diabetic NOD donor or vehicle control. Animals were scanned again 2 weeks later ("2w post"), followed by I.P. injection of 20 μg anti-CD4 the following day for those receiving antiCD4 treatment. Animals were scanned again 2 weeks later ("4w post"). All AT animals were monitored weekly for weight and

glucose until animals presented with consecutive weekly glucose measurements of >250 mg/dl. Each animal acted as its own internal control, and comparisons are made within individual animals, compared to CEUS scan before transfer. Comparisons were also made between all experimental groups of animals.

**Histology or insulitis and vascular morphology.** For assessment of insulitis, all animals were anesthetized by I.P. injection of ketamine (80 mg/kg) and xylazine (16 mg/kg) until no longer reactive to toe pinch; the pancreata were dissected and mice were euthanized. Pancreata were fixed in paraformaldehyde at 4 °C rocking overnight and embedded in paraffin blocks. Five micron sections from at least three different tissue depths were stained for H&E to evaluate islet monocyte infiltration. Islets were scored based on the extent of infiltration/insulitis: grade 0, no insulitis; grade 1, peri-insulitis with immune infiltrate bordering, but not entering islet structure; grade 2, immune infiltrate penetrating the islet, covering <50% of islet area; and grade 3, immune infiltrate penetrating the islet, covering >50% of islet area. A minimum of three different tissue depths and at least 50 nonoverlapping islets per animal were analyzed. Weighted averages were calculated for each animal. Images were acquired on an Eclipse-Ti wide field microscope with a ×20 0.75 NA Plan Apo objective with a color CCD camera.

For assessment of morphological changes in islet vasculature, female NOD mice (JAX) were anesthetized by I.P. injection of ketamine (40 mg/kg) and xylazine (8 mg/kg). Following anesthesia, 100 μg of Texas Red labeled *Lycopersicon esculentum* (tomato) lectin ("TL"; Vector Laboratories, #TL1176, Burlingame, CA) was injected in a single bolus into the lateral tail vein, similar to previously described[30]. The TL was allowed to circulate for 5 min, followed by another injection of ketamine (40 mg/kg) and xylazine (8 mg/kg) to induce deep anesthesia. Following pancreas isolation, tissues were fixed in 4% PFA on ice for 1 h and cryoprotected in 30% sucrose overnight or until the tissue sank. The TL-infused pancreata were embedded in OCT medium, frozen in cryomolds, and sectioned at 10–20 μm. Sections were imaged at 595 nm excitation on an LSM800 confocal microscope (Zeiss), using 1 μm optical sections throughout the tissue depth. Separate images were taken of exocrine tissue at locations anatomically isolated from endocrine islets. The TL coverage was calculated in MATLAB as percent positive pixels across the islet (background subtracted), and expressed as a fraction of the islet area. The microvascular diameter was determined by manual measurement of TL-positive vasculature, and averaged across all imaged islets for individual animals. Both vascular quantification measures (coverage and diameter) were compared to vasculature in pancreatic exocrine tissue. The image analyzer was blinded to experimental groups under analysis.

**Statistical analysis.** All data are presented as mean ± SEM. Statistical comparisons were made using paired or unpaired Student's $t$-tests or ANOVAs, wherever appropriate and as indicated. Where multiple measurements per mouse were included, a mixed-effects model ('fitlme()' in MATLAB or PROC MIXED in SAS) was employed. Outliers, as defined by Grubbs' test, were excluded for the purpose of statistical analysis, but retained in figure panels. Statistical significance was taken as $p < 0.05$. A Bonferroni correction for multiple comparisons were performed where indicated.

When testing for disease prediction, receiver operating characteristics (ROC) curves were generated where sensitivity was defined as percentage of NOD mice or AT mice within some disease threshold and 1-specificty was defined as percentage of NOD-RAG1⁻/⁻ mice or vehicle control mice within some disease threshold. Maximum likelihood analysis was employed to define an optimum disease prediction threshold. This threshold was then employed to separate either NOD mice at 12 weeks or antiCD4-treated AT mice (i.e., a separate cohort of animals) into disease positive and disease negative groups, which were subsequently compared.

Sample sizes for experimental groups were based on the results of the STZ study to provide sufficient statistical power, given the measured-effect size (difference in reperfusion parameters). When comparing experimental groups, CEUS recordings were not made in a defined order.

**Data availability.** All datasets generated during the current study are available from the corresponding author on reasonable request.

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

## Acknowledgements

We thank Mark Borden and Jane Reusch (University of Colorado) for helpful discussion and Laura Pyle (University of Colorado) for assistance with statistical analysis. R.K.P.B. (University of Colorado) is the guarantor of this work and, as such, had full access to all the data in the study and takes responsibility for the integrity of the data and the accuracy of the data analysis. This work was supported by Juvenile Diabetes Research Foundation Grants 5-CDA-2014-198-A-N, 1-INO-2017-435-A-N; and NIH grants R01 DK102950, R01 DK106412, U01 AI101990 subaward (to R.K.P.B.) and NIH grants T32 HL072738, F32 DK112525 (to J.R.S.).

## Author contributions

J.R.S. designed and performed experiments, analyzed data, wrote the manuscript; D.R. performed experiments, analyzed data; S.P. analyzed data; R.K.P.B. designed experiments and wrote the manuscript.

## Additional information

**Competing interests:** The authors declare no competing interests.

