## [Peer Review File · Nature Communications]

Reviewers' comments:

Reviewer #1 (Remarks to the Author):

General

The authors investigate the use of contrast-enhanced ultrasound measurements (CEUS) of pancreas blood-flow dynamics in pre-clinical murine models to measure progression of diabetes. There is clearly a critical unmet medical need in diabetes to develop non-invasive method to access both beta-cell mass and monitor response to treatment. The authors validate CEUS by performing intraperitoneal glucose test and an insulin tolerance test on a non-diabetic mouse strain. They then move on to induce hyperglycaemia by streptozotocin (STZ) treatment at two doses and, measure the effect on pancreatic perfusion. Finally, they switch to the NOD model of autoimmune diabetes and a NOD-RAG1 knockout transfer model where they measure pancreatic perfusion at several longitudinal time-points with and without immunotherapy. On analysis of their preclinical results they conclude that CEUS could be translated to the monitor and predict type 1 diabetes (T1D) in patients.

Title

The title needs to make clear that the studies have been carried out in murine models of diabetes, not type 1 diabetes.

Validity

Though I am not expert in ultrasound, the validation experiments look reasonable. The pre-clinical models also seem appropriate, though there is little discussion of their each of their limitations and, why different models were employed. However, I am concerned over the confounding effect of glucose on their measurement of vascular changes. Though they have attempted in some respects to address this, it is not clear to me that the vascular changes measured are due to inflammation/autoimmunity or just hyper/hypoglycaemia. Since the pathology of type 2 diabetes (T2D) differs from that T1D, the inclusion of a mouse model of T2 diabetes in the studies would assist with the interpretation and the generalisability of the results.

Originality and significance

Despite the limitations, this is a well conducted study to develop a non-invasive method to measure pancreatic blood flow to monitor beta cell loss in mice during onset of diabetes. The post-hoc analysis to predict the development of diabetes, is not compelling. If the authors wish to make this claim, then they need to prospectively carry out new study using the reperfusion rate that they have derived.

Data and Methodology

The data and methodology are well described, it would be nice to know the durations of general anaesthesia and procedures which are relevant to the transability of CEUS.

Statistics

There are multiple tests so a statistical level $P < 0.05$ will result in false positives. It is unclear to me how the reperfusion threshold of 0.8 was derived to predict diabetes.

Conclusions

Overall the conclusions are too long, the study is preclinical, its findings are limited to the models investigated, while they may have some relevance to patients this has not been shown yet, so the claims of relevance should be limited.

Specific

"Therefore, the increased perfusion velocity in STZ treated animals indicates that CEUS can non-invasively detect changes in islet blood flow dynamics in vivo associated with β -cell injury and islet inflammation"

It's unclear if this is due to inflammation

Therefore the increased perfusion velocity in the AT model can non-invasively detect

disease progression and strongly predict disease reversal upon therapeutic treatment, further suggesting a diagnostic potential.

This is overreach, as a new prospective study is needed

Specific minor comments

Introduction

"At disease onset most, but not all, of the β -cells have been destroyed, thus limiting successful therapeutic intervention"

This is a historical review, newer literature post 2015 needs to be consulted

"cannot be correlated with the underlying disease progression"

In humans, autoantibodies can be correlated with autoimmune disease progression

"Furthermore, decreases in islet vascular density and compensatory increases in vessel diameter have been observed in mouse models of type2 diabetes (T2D), and humanT2D donors"

See my early comments about the use of a T2D model as controls

Results

"Given that pancreatic blood flow is predominantly governed by islet blood flow"

This conflicts with later statements in the discussion, and is not clinically plausible

"However, and somewhat unsurprisingly, there was a modest correlation of reperfusion rate with animal heart rate."

Given the number of correlation tests, this may be a false positive

"The non-obese diabetic (NOD) mouse develops spontaneous autoimmune diabetes as a result of infiltration of autoreactive T-cells into the islets of Langerhans many weeks before diabetes onset; similar to human patients with T1D albeit with different time scales and intensity"

The insulinitis in the NOD differs from humans, the sentence needs to be revised to reflect this.

"However, there is no reliable way to identify or monitor treatment efficacy".

This is not entirely true, again it needs to be changed.

"We utilized an adoptive transfer (AT) model of inducible diabetes, where delivery of donor splenocytes lead to well-defined T-cell mediated autoimmune diabetes in the recipient"

Unclear why this model was used rather than the NOD

"We again performed post-hoc analysis to examine how well CEUS measurements could predict diabetes development"

Post-hoc analysis is not adequate for the claims made.

Discussion

"There are no reliable and reproducible approaches to diagnose T1D during the pre-symptomatic phase of disease progression, given the lack of clinical presentation of symptoms until significant β -cell death has occurred, heterogeneity in underlying disease progression, and a technical inability to non-invasively monitor β -cell decline and insulinitis over time."

There are no non-invasive methods, there are invasive methods, revise.

"Therefore, CEUS measurements of pancreas blood flow dynamics provides a means to non-invasively and longitudinally track the underlying disease to the islets of Langerhans with high sensitivity and specificity, and predictive of T1D presentation."

Not predictive of T1D as this disease only occurs in humans, predictive of diabetes in the models studied, revise.

"Nevertheless we anticipate even a small number of longitudinal 'snap shots' will provide useful for pre-clinical studies to monitor the effectiveness of potential therapeutic treatments"

What type of patient studies? expand

"Further while robust decreases in reperfusion amplitude were only observed in the NOD model, this is consistent with a longer duration of disease development compared to AT and STZ models, suggesting the decline in islet microvasculature coverage is a slow process and relevant to human T1D"

The progression of T1D dependent on the age of the patients, faster in the young and slower in older patients, revise

"The endocrine compartment receives approximately one fifth of the total pancreatic blood supply despite constituting 1-2% of pancreas volume"

See comment above in results, this conflicts with earlier statement

"Furthermore, acute increases or decreases to glucose only affect islet blood flow with no effect on exocrine blood flow and it is under these manipulations that we observed changes in blood flow dynamics similar in magnitude to those observed in the models of T1D."

This is my main concern, that glucose caused the vascular effects.

"Data examining the islet microvasculature in human T1D is lacking. However given similarities in islet microvasculature changes between mouse and human and between T1D and T2D we anticipate that CEUS would be applicable to pre symptomatic human T1D."

This is overreach, as there is no data in T1D.

"The pace of disease progression is also slower in humans than what is observed in NOD mice, estimated to occur over many years with a lower intensity of insulinitis."

The rate progression of T1D in patients is unclear other than the effect of age.

"Of note the CEUS approach may also be translated to other diseases involving localized inflammatory responses, such as multiple sclerosis."

Difficult to see how transcranial ultrasound could be used in a similar manner, revise.

Reviewer #2 (Remarks to the Author):

In their manuscript, St. Clair, Benninger and colleagues use a non-invasive ultrasound approach to study changes in pancreatic blood flow during type 1 diabetes progression in mice. The authors test the novel and brilliant idea that changes in islet inflammation associated with the development of diabetes can be monitored non-invasively by measuring pancreas blood flow. The premise makes perfect sense: inflammatory conditions increase tissue perfusion and islets have a relatively large blood supply within the mouse pancreas. The results showing that pancreatic blood flow measurements can predict diabetes in different mouse models are very convincing and support the conclusions of the paper. There is a sound validation of the technique; the authors establish reproducibility and consistency and provide all the necessary controls. The changes in blood flow are impressive and indeed predict diabetes progression in mice. The discussion is thorough and addresses shortcomings as well as the future applicability of the method in human subjects. After reading the paper, the reader is left wondering why this simple but powerful approach has not been tested before. The obvious next step is to implement this technique in human beings, or at least in primate models.

There are some issues that should be addressed to make this fine manuscript even stronger.

Issue 1: The main issue is whether or not the approach transfers well to the human situation. It is encouraging that ultrasound measurements of pancreatic blood flow have already been used in human subjects using a similar technique, as referenced by the authors. It is also clear that there are substantial changes in the human islet vasculature in diabetes. What is not clear is what proportion of the pancreas blood flow readout will be representative of islet blood supply. The vasculature of the human islet is almost continuous with that of the exocrine tissue and does not show the higher vascular density typical of mouse islets (Brissova et al., 2015, *J Histochem & Cytochem* 63:637; Cohrs et al., 2017, *Endocrinology* 158:1373). If the contribution of islet blood flow to total pancreas blood flow is too small in the human pancreas, changes occurring during diabetes progression may remain undetected. This study cannot address this issue, but this potential limitation has to be addressed and discussed.

Issue 2: It is known that anesthetics including isoflurane can change glycemia by reducing insulin secretion. Was this controlled for in the measurements? Because not all animals respond equally with changes in glycemia to anesthesia, this may account partially for some of the variability.

Issue 3: It is important to use appropriate statistical tests when multiple comparisons are made. Student's t tests were used inappropriately in several of the figures (e.g. Figure 3B, C, and H).

Minor points:

Line 49: Please mention the approaches that hold promise.

Line 124, Figure 1D: how do these changes compare to what has been shown before with more invasive techniques?

Line 218: What can explain the increase in vessel diameter in NOD mice before the onset of diabetes? Inflammation? What does the literature say? Some discussion is needed here.

Line 243: For this pharmacological intervention, please cite references using this protocol

Line 251: Authors mean Figure 5E, not 5F

Line 280: Stating that a "multitude" of models were used is a stretch because two and a half models were used.

Reviewer #3 (Remarks to the Author):

St Clair et al report on the use of contrast-enhanced ultrasound (CEUS) measurement of pancreatic blood flow dynamics in mice as predictor of diabetes progression and therapeutic reversal. This reviewer finds this paper to be an exciting and important concept with significant potential for translation into humans, and hopes that the authors will be able to address the issues raised.

Strengths:

1. While there is a recent paper describing the use of ultrasound in BB rat model of diabetes (PLoS ONE 12(6): e0178641.), the use of CEUS to image pancreatic inflammation is novel and there is an unmet need to "see" what is occurring in the pancreas as a predictor of disease progression, to evaluate response to therapy aimed at beta cell preservation, and to better understand the pathophysiology of disease in human type 1 diabetes.
2. There has been no real success in imaging pancreatic beta cells themselves, thus the idea of imaging "inflammation". In the current pre-clinical study, the investigators evaluate the utility of whole pancreas blood flow reperfusion rate and amplitude by CEUS. Whether the sensitivity and specificity seen in small animals can be translated to humans without employing additional imaging modalities for isolating the pancreas for transcutaneous US will need to be tested. Nonetheless, CEUS has significant advantages for future clinical evaluation and use since the lack of radiation and likely very high safety suggests it could be used for longitudinal studies as well as in pediatric age groups.

Weaknesses

In addition to logistical and safety considerations, key elements for translation of new tools for diagnostic or prognostic evaluation in humans depend on sensitivity and specificity, variance in populations, predictive values, and within subject reproducibility among others. The investigators have performed multiple assessments to address many of these elements, but their conclusions would be strengthened by addressing the following items:

1. Reproducibility and variance data:
 - a. Reproducibility and variance are important to translation. Thus, the authors should consider moving components of the figures from supplement to the main paper.
 - b. (S1A-C) What is the explanation for the wide variance in the C57BL6/J mice in both measures of rate and amplitude? These are genetically identical mice who presumably have no insulinitis or abnormal glucose. Moreover, mean and range of amplitude in both NOD and NOD-Scid mice (also without insulinitis or abnormal glucose) appear quite similar. Explain.
 - c. (S1D,E) Reproducibility over two visits should be shown for each individual mouse. The time between the tests should be shown. A T test is not appropriate for these comparisons. Moreover, there is significant variability in response of amplitude to glucose (Fig 1D) that is not considered. Text indicates no association with "day/time, reagents or operators", yet this is not clear from legend or methods section and no data is shown. What is "normalized" amplitude and why is that used in this figure and no other?
2. Figure 1 D, E
 - a. This experiment was done to determine the effect of increasing (GTT) or decreasing (ITT) glucose values on the blood flow measures. The results should be presented to make the relationship clear (i.e. each animal glucose on one axis and blood value measures on the other).
 - b. It would be useful to know if the animals undergoing GTT were the same as those that underwent ITT; while the baseline glucose values appear similar, there are differences in baseline rate and amplitude measures. This raises the question as to whether the differences in baseline measures exceed the differences from GTT or ITT testing and again points to questions about variability in measures in the same animal over time.
3. Figure 2
 - a. With the small N, it would be best to show individual data in 2A, and to connect the lines both in STZ and control animals (2B,C). It would also be useful to plot the glucose value at time of

measurement of blood flow.

b. Assure that the N in the figure matches the N in the legend

c. The authors assert that STZ (50) represents animals without "overt hyperglycemia"; and that the animals are "euglycemic" except for the glucose intolerance suggesting that this represents a pre-diabetic state. There is no evidence that the animals are "euglycemic" particularly given the dramatic hyperglycemia to the GTT in many. The authors also consider that there is an increase in rate in these animals; however, this difference is clearly limited to 2 animals – which, since the authors note that glucose and blood flow are related, could be due to hyperglycemia. These data do not support the idea that the rate is increased during "pre-clinical" period.

4. Figure 3

a. Figure 3 consists of analysis in NOD and NOD Rag1^{-/-} mice to illustrate the relationship of blood flow dynamics with disease progression. As evident from 3A, NOD animals develop hyperglycemia at different time points – with some without progression. As such, presenting the group data at selected time points (3B, C) is not particularly helpful. More useful is 3D – showing individual changes over time; however, this plot should also include 18 week data and a similar plot should be done for the amplitude data since the authors suggest elsewhere that an increase and rate and decrease in amplitude both occur during disease progression. Importantly however, since glucose and blood flow are related, showing individual animal blood flow data over time along with the glucose over time would allow the reader to better understand the relationship of blood flow measures with disease progression (at least as measured by these random glucose values). Similarly, the group data (3E,F) are not as useful as the longitudinal data (3G), which should include the 18 week time point.

b. The authors use the NOD Rag1^{-/-} mice as controls. While these animals do not develop diabetes, another control to strengthen the authors' conclusions, with an intact immune system, to consider are NOD male animals or showing longitudinal data in the C57BL6/J mice to provide information about how these variables change with age over time in health.

c. 3H depicts "steady-state" intensity for which, while noted in methods, there is no description in the legend. It is not clear why this is an important measure. Are the authors suggesting that "steady-state" analysis provides additional information to the dynamic measures that have predictive or mechanistic value? Moreover, this graph as well as 3K present bar plots and means when individual longitudinal data would be more informative. Separately, the N for 3H is not provided in the legend. If same as for other figures, there are 23 NOD and 6 NOD Rag1^{-/-}. From the lack of the asterisk, the authors imply no changes in this measure over time in the NOD Rag1^{-/-} (also line 182); but the small number of animals makes this an uncertain interpretation.

d. Figures 3I, J uses all data points from 23 mice (62 scans). Thus, these are repeated measures. A different analytic approach should be used addressing this issue and the graphs should indicate the individual mice.

e. Figure 3K, data better shown as individual dots rather than bar graphs. Moreover, figure legend indicates the parameters were those that progressed to diabetes < and > 10 weeks, while the figure itself uses > and <5 weeks. In neither case is the = clear. There is no explanation as to why 5 and 10 weeks were chosen.

5. Figure 4

a. The authors group data from a total of 49 islets from 10 mice. This is an insufficient number of islets examined to make conclusions about changes in vessels from 5 to 10 weeks. Moreover, as the authors illustrate previously, there is wide variability in disease progression in individual animals at these time points. The vascular measures from each animal at the time points should be associated with the CEUS measures from those animals at those time points.

6. Figure 5

a. It is not clear why the authors use anti-CD4 in the AT model as the treatment to demonstrate effects of therapy when there is literature using anti-CD3 in NOD mice and humans.

b. 5A-C. The legend indicates 11 AT mice and 10 controls; however, the number of dots plotted, particularly for controls is less. As before, connecting the dots to demonstrate individual animals over time would be helpful, as well as associating the glucose results with the values since these is a large amount of variability in rate at 2 and 4 weeks in the AT animals.

c. 5D-F. With only a total of 14 mice – individual data would be useful rather than mean changes.

Moreover, the definition of non-responder are those that became hyperglycemic; it appears as if 1 or 2 of the non-responders may have been hyperglycemic at the time of infusion and thus, their data may not reflect (non-)response to therapy. Moreover, interpretation of this data is highly dependent upon day to day reproducibility without interventions, which as noted above (comment 1A), is not well presented.

7. Figure 6

a. This illustration is not needed for the paper. Moreover, it suggests that disease progression is associated with increased rate and decreased perfusion; yet this is not consistent with all of the data shown. For example, no differences in amplitude in AT mice at week 4 despite many developing diabetes (Figure 5C); an inconsistent amplitude response to GTT (1D), no effect on amplitude in STZ treated animals (2C,F).

8. Table 1

a. This table aims to illustrate a predictive value for development of diabetes. As noted above, there is no rationale given for the 5 and 10 week time periods (and the categories exclude animals who develop diabetes at 5 and 10 weeks). There is no information provided as to how the value of >0.8 was derived. The analysis gives no information as to the number of measurements in each animal, nor the variance in the measures. If the authors believe that there is a rate value that predicts time to diabetes, it should be a rate value at a given time and not a group of measures over time. For example, what is the positive predictive value of a rate of X for development of diabetes? The authors should consider use of ROC curves and models that predict categorical yes/no diabetes as well as time to diabetes. Alternatively, the authors could present the data (values) over time in those who do and do not progress so that the variability in this can be readily observed in all animals.

9. Figure S4 (A-C)

a. Legend should clarify what "weighted" average means and, since only 5 mice whether there were differences in the number of islets measured in each mouse (legend states "minimum of 60 islets"). The title of the figure indicates that the insulinitis "correlates" with reperfusion rate; yet the data is not presented that way. The data should illustrate the relationship of the individual animal's insulinitis score with their reperfusion rate at 12 weeks. Moreover, the legend says that S4C is the amplitude and the figure label says this is the rate. Both variables should be shown.

10. Other comments

a. Discussion; lines 304-306. Authors suggest that two independent parameters would be more specific and predictive. Yet, table 1 only considers one parameter.

b. Discussion: lines 323-325. The authors could use their data to test this statement.

c. Discussion: a recent publication (Roberts FR et al. PLoS ONE 12(6): e0178641) describes the use of US in BB rat and should be considered in the discussion.

Below follows each of the reviewers comments (*italics*) and our detailed response to each comment, together with a description of where and how we have modified the manuscript. Where relevant we include numbers for specific reviewer comments to refer to later on. In general we agree with the vast majority of the reviewers comments and believe addressing their suggestions or concerns has substantially improved the manuscript. All changes in the manuscript are indicated in **red** and we underline the position in the manuscript we make relevant changes.

Reviewer #1

1. General

The authors investigate the use of contrast-enhanced ultrasound measurements (CEUS) of pancreas blood-flow dynamics in pre-clinical murine models to measure progression of diabetes. There is clearly a critical unmet medical need in diabetes to develop non-invasive method to access both beta-cell mass and monitor response to treatment. The authors validate CEUS by performing intraperitoneal glucose test and an insulin tolerance test on a non-diabetic mouse strain. They then move on to induce hyperglycaemia by streptozotocin (STZ) treatment at two doses and, measure the effect on pancreatic perfusion. Finally, they switch to the NOD model of autoimmune diabetes and a NOD-RAG1 knockout transfer model where they measure pancreatic perfusion at several longitudinal time-points with and without immunotherapy. On analysis of their preclinical results they conclude that CEUS could be translated to the monitor and predict type 1 diabetes (T1D) in patients.

We appreciate the reviewer indicates that “There is clearly a critical unmet medical need in diabetes to develop non-invasive method to access both beta-cell mass and monitor response to treatment”

2. Title

The title needs to make clear that the studies have been carried out in murine models of diabetes, not type 1 diabetes.

We have modified the title to indicate we are referring to murine models of T1D

3. Validity

Though I am not expert in ultrasound, the validation experiments look reasonable. The pre-clinical models also seem appropriate, though there is little discussion of their each of their limitations and, why different models where employed. However, I am concerned over the confounding effect of glucose on their measurement of vascular changes. Though they have attempted in some respects to address this, it is not clear to me that the vascular changes measured are due to inflammation/autoimmunity or just hyper/hypoglycaemia. Since the pathology of type 2 diabetes (T2D) differs from that T1D, the inclusion of a mouse model of T2 diabetes in the studies would assist with the interpretation and the generalisability of the results.

We have included a brief description for why specific models were included – see Results page 6, paragraph 2 (4th line); Results page 6, paragraph 5 (3rd line); Results page 9, paragraph 1 (4th line).

We also acknowledge that each model has some limitations in modelling human T1D – see Discussion page 13 paragraph 3 (9th line).

We agree that changes in glycemia have the potentially to confound our results. However we do note that in nearly all models where we measure pancreas blood flow dynamics the mice were euglycemic at the time of measurement. This includes NOD mice at 6 and 12 weeks, adoptive transfer mice 2 and 4 weeks following splenocyte delivery, and low (50mg/kg) STZ – this is summarized below, and we include a new supplemental figure showing this - see Figure S4. While higher dose STZ (70mg/kg) were hyperglycemic, similar measurements were observed with 70mg/kg STZ (hyperglycemic) and 50mg/kg STZ (euglycemic, glucose intolerant). Further while NOD mice at 18 weeks showed a small elevation in glucose compared to 12 weeks, we also observed similar blood

flow measurements at each time point. Finally, to also exclude an association with glucose intolerance, while lower dose 50mg/kg STZ are glucose intolerant compared to vehicle controls, NOD mice at 12 weeks show only minor change compared to Rag1ko controls (below). Thus we conclude that even glucose intolerance cannot explain blood flow changes in NOD mice.

We have previously measured a cohort of db/db animals prior to diabetes and during/after their development of diabetes. Despite these animals progressing to severe hyperglycemia, we did not observe a substantial change in pancreas blood flow dynamics. While there is some trend, this only occurs at 8 weeks when naturally some inflammation at the islet may be expected, and the changes are much less than in STZ, NOD and AT animals. We include this data in supplemental figure S7 and mention this data in Results – see page 7 paragraph 3 to page 8 paragraph 1.

We have also performed optogenetic stimulation specifically of pancreas parasympathetic innervation and also observe significant increases in blood flow (below), although the method used to achieve this is beyond the scope of this paper and thus we only include here to show glucose and diabetes independent regulation

Above: summary of blood glucose levels and reperfusion rate in all groups examined. Note in many cases large changes in reperfusion rate occur despite no changes in blood glucose, and further when blood glucose is elevated there is no change in reperfusion rate. (where in F the elevation is skewed by 2 animals that were hyperglycemic)

[Redacted]

Above: db/db (blue) and C57BLKS control (black) blood glucose and pancreas reperfusion dynamics at ages indicated

[Redacted]

4. Originality and significance

Despite the limitations, this is a well conducted study to develop a non-invasive method to measure pancreatic blood flow to monitor beta cell loss in mice during onset of diabetes. The post-hoc analysis to predict the development of diabetes, is not compelling. If the authors wish to make this claim, then they need to prospectively carry out new study using the reperfusion rate that they have derived.

We thank the reviewer for noting this is a well conducted study. As discussed and addressed below (see comment on conclusions), we acknowledge this study in preclinical models does not demonstrate validity in humans, only suggests the possibility. However given the ease of which CEUS could be translated, we think our study does provide a significant impact. Further we also recognize the reviewer's point regarding the posthoc analysis. We have removed table1 and associated text. We simply state that 3/3 responders but only 3/9 non-responders mice with a rate that was greater than the highest quartile level observed in control mice progress to diabetes – see page 9 paragraph 2 (14th line).

We also note that to further test the ability to predict drug success that prospective study would be needed, and this will be a goal for future work for a number of immunotherapies or beta cell therapies – see Discussion page 11 paragraph 1 (23rd line).

5. Data and Methodology

The data and methodology are well described, it would be nice to know the durations of general anaesthesia and procedures which are relevant to the transability of CEUS.

Anesthesia was for the duration of the scan, which from initial application of isoflurane until the end of the scan was 15-20 minutes for all mouse strains (B6 mice, NOD, NOD-Rag1ko and Nod-scid mice). We now include this detail – see Methods page 15 paragraph 2 (1st line).

Elsewhere in the Methods we endeavor to include as many specific details as possible, e.g. infusion set, VEVO 2100 settings, Size-isolated microbubble delivery.

6. Statistics

There are multiple tests so a statistical level $P < 0.05$ will result in false positives. It is unclear to me how the reperfusion threshold of 0.8 was derived to predict diabetes.

We understand the confusion as in figure captions we only state t-tests – this is because we only compared animals in the same group between early and late in disease. We did not for example compare STZ-treated with control treated. However following reviewer 3 comments, we have modified our figures such that where multiple groups are presented an ANOVA is used, but where we compare a single group at 2 time points a paired t-test is used. In each figure caption, we now indicate which comparisons utilized a t-test and which comparisons utilized an ANOVA – see Captions pages 25-27.

With respect to the reperfusion threshold of 0.8, this was based on the highest rate measured in control mice during the initial AT experiments (equivalent to mean+2SD or 95th percentile). As such it was chosen independent of NOD or Rag1ko measurements and independent of antiCD4 treated AT measurements. However, following the comment above we have removed this table and associated text.

7. Conclusions

Overall the conclusions are too long, the study is preclinical, its findings are limited to the models investigated, while they may have some relevance to patients this has not been shown yet, so the claims of relevance should be limited.

We have reduced the length of the discussion from ~1900 words to ~1500 words, which includes addressing this reviewer's and other reviewers' comments.

We agree with the reviewer that our findings do not demonstrate immediate translatability for detecting human pre-symptomatic T1D. Our conclusions are limited to the preclinical models utilized, but together with considerations of the islet microvasculature in human T1D and the utility of contrast enhanced ultrasound, there may be translatability to patients and our study indicates the need to test this. We note reviewers 2 and 3 noted the possibility for clinical translation. But of course a clinical study would be needed to validate this. We have therefore reworded several sections throughout the discussion to reflect this, including Discussion page 11 paragraph 1 (21st line); page 13 paragraph 2 (14th line).

8. Specific

“Therefore, the increased perfusion velocity in STZ treated animals indicates that CEUS can non-invasively detect changes in islet blood flow dynamics in vivo associated with β -cell injury and islet inflammation”

It's unclear if this is due to inflammation

We absolutely agree with this comment and apologize for this oversight. Throughout agree that while inflammation is associated with each model employed, we do not know whether this caused the altered blood flow dynamics. We have removed this statement, but at the suggestion of review 2 we mention how it remains to be determined whether there is a link between blood flow and inflammation in the islet – see Discussion page 13 paragraph 1 (2nd line).

Therefore the increased perfusion velocity in the AT model can non-invasively detect disease progression and strongly predict disease reversal upon therapeutic treatment, further suggesting a diagnostic potential.

This is overreach, as a new prospective study is needed

We agree with this comment, and have removed this statement, and now mention how a prospective study is required – see Discussion page 11 paragraph 1 (23rd line).

9. Specific minor comments

Introduction

“At disease onset most, but not all, of the β -cells have been destroyed, thus limiting successful therapeutic intervention”

This is a historical review, newer literature post 2015 needs to be consulted

Thank you. Indeed, this is indicated in a number of reviews, including more recent ones post 2015 (e.g. A.G Ziegler et al Diabetes 2016,). However while widely stated and perhaps obvious, upon a deeper reading there is not actually too much solid clinical data supporting this statement (but there are some studies). We thank the reviewer for their prodding at this, and as such we add this more recent literature but also modify the statement. – see Introduction page 3 paragraph 1 (8th line).

“cannot be correlated with the underlying disease progression”

In humans, autoantibodies can be correlated with autoimmune disease progression

Thank you – we instead say “do not reflect the potential to therapeutically reverse the underlying disease – See Introduction page 3 paragraph 1 (10th line).”

“Furthermore, decreases in islet vascular density and compensatory increases in vessel diameter have been observed in mouse models of type2 diabetes (T2D), and human T2D donors”

See my early comments about the use of a T2D model as controls

See our response to point 3.

Results

“Given that pancreatic blood flow is predominantly governed by islet blood flow”

This conflicts with later statements in the discussion, and is not clinically plausible

We have removed this statement. Rather the pancreas blood flow correlates with islet blood flow. To highlight this rationale for measuring pancreas blood flow we state this in the Introduction – see page 3 paragraph 3 (12th line); and elaborate upon it in the Discussion – see page 13 paragraph 2 (1st line).

“However, and somewhat unsurprisingly, there was a modest correlation of reperfusion rate with animal heart rate.”

Given the number of correlation tests, this may be a false positive

We have performed a multiple comparisons correction to account for false positives associated with comparing a set of data against multiple parameters. This still reveals that the association between reperfusion rate – heart rate is statistically significant. This does not affect any of our conclusions however. We include the details of this correction within figure S2.

“The non-obese diabetic (NOD) mouse develops spontaneous autoimmune diabetes as a result of infiltration of autoreactive T-cells into the islets of Langerhans many weeks before diabetes onset; similar to human patients with T1D albeit with different time scales and intensity”

The insulinitis in the NOD differs from humans, the sentence needs to be revised to reflect this.

We remove the part of this statement that refers to comparison with human patients. We already state in the introduction that in T1D there is immune cell infiltration of the islet. We believe it sufficient to state that since the NOD mouse shows islet infiltration by autoreactive T-cells and thus models T1D better than STZ injury– see Results page 6 paragraph 5 (2nd line).

*“However, there is no reliable way to identify or monitor treatment efficacy”.
This is not entirely true, again it needs to be changed.*

We agree with this since some aspects of glycemia or c-peptide response may provide some indication. Instead we say “there are limited means to monitor treatment efficacy” – see Results page 9 paragraph 1 (2nd line).

*“We utilized an adoptive transfer (AT) model of inducible diabetes, where delivery of donor splenocytes lead to well-defined T-cell mediated autoimmune diabetes in the recipient”
Unclear why this model was used rather than the NOD*

Only a subset of NOD females progress to diabetes (~80%), and progression to diabetes occurs over a long duration (~40 weeks to reach ~80%). Thus to test for an association of blood flow changes with diabetes prevention prior to onset, in a situation where only a subset are protected, a large numbers of scans covering almost 1 year would be needed, which is not feasible. Instead, 100% of adoptive transfer animals develop diabetes within 6 weeks. Thus fewer animals and a shorter experimental duration is required to test for an association between blood flow measurements and development of diabetes. We now include a brief motivation for utilizing the AT model to assess therapeutic reversal – see Results page 9 paragraph 1 (4th line).

*“We again performed post-hoc analysis to examine how well CEUS measurements could predict diabetes development”
Post-hoc analysis is not adequate for the claims made.*

As discussed above in response to point 4 we have removed the posthoc analysis

Discussion

“There are no reliable and reproducible approaches to diagnose T1D during the pre-symptomatic phase of disease progression, given the lack of clinical presentation of symptoms until significant β -cell death has occurred, heterogeneity in underlying disease progression, and a technical inability to non-invasively monitor β -cell decline and insulinitis over time.”

There are no non-invasive methods, there are invasive methods, revise.

We intended to say non-invasive to distinguish from autopsy based assessments, although this is perhaps obvious. Thus we remove ‘non-invasive’ – see discussion page 11 paragraph 1 (4th line).

“Therefore, CEUS measurements of pancreas blood flow dynamics provides a means to non-invasively and longitudinally track the underlying disease to the islets of Langerhans with high sensitivity and specificity, and predictive of T1D presentation.”

Not predictive of T1D as this disease only occurs in humans, predictive of diabetes in the models studied, revise.

As in response to point 7, we state here and elsewhere that the tracking of T1D or prediction of T1D onset is in the models employed here, e.g. – see Discussion page 11 paragraph 1 (21st line); page 13 paragraph 2 (14th line).

“Nevertheless we anticipate even a small number of longitudinal ‘snap shots’ will provide useful for pre-clinical studies to monitor the effectiveness of potential therapeutic treatments”

What type of patient studies? expand

We refer to disease prevention studies where therapeutic intervention occurs prior to disease onset during the presymptomatic phase. We expand on this – see discussion page 12 paragraph 2 (11th line).

“Further while robust decreases in reperfusion amplitude were only observed in the NOD model, this is consistent with a longer duration of disease development compared to AT and STZ models, suggesting the decline in islet microvasculature coverage is a slow process and relevant to human T1D”

The progression of T1D dependent on the age of the patients, faster in the young and slower in older patients, revise

We appreciate this statement is perhaps too speculative and have removed it.

“The endocrine compartment receives approximately one fifth of the total pancreatic blood supply despite constituting 1-2% of pancreas volume”

See comment above in results, this conflicts with earlier statement

While this statement is correct (but as pancreas blood flow correlates with islet blood flow) we have removed this statement in the course of reducing the length of the discussion

“Furthermore, acute increases or decreases to glucose only affect islet blood flow with no effect on exocrine blood flow and it is under these manipulations that we observed changes in blood flow dynamics similar in magnitude to those observed in the models of T1D.”

This is my main concern, that glucose caused the vascular effects.

We understand the importance of this point. See our discussion above in response to point 3. Briefly, to reiterate, NOD and AT mice were euglycemic, and NOD mice show only minor changes in glucose tolerance at the point in which changes in blood flow associated with disease were observed. Further, db/db animals do not show such changes despite being chronically hyperglycemic. Thus while glucose changes could cause vascular changes, our data indicates they are not associated with the changes observed here. We now state the glucose data – see Results page 6 paragraph 4 (2nd line), page 8 paragraph 2 (3rd line), and Discussion page 13 paragraph 2 (6th line).

“Data examining the islet microvasculature in human T1D is lacking. However given similarities in islet microvasculature changes between mouse and human and between T1D and T2D we anticipate that CEUS would be applicable to pre symptomatic human T1D.”

This is overreach, as there is no data in T1D.

We agree – we now simply state the data regarding mouse models of T2D and human T2D, and that measurements of microvascular permeability in mouse and human T1D are similar. We are also aware of a study in review indicating changes in established human T1D and refer to this – see Discussion page 13 paragraph 3 (13th line).

“The pace of disease progression is also slower in humans than what is observed in NOD mice, estimated to occur over many years with a lower intensity of insulinitis.”

The rate progression of T1D in patients is unclear other than the effect of age.

We have removed this statement.

*“Of note the CEUS approach may also be translated to other diseases involving localized inflammatory responses, such as multiple sclerosis.”
Difficult to see how transcranial ultrasound could be used in a similar manner, revise.*

We have removed this statement.

Reviewer #2:

In their manuscript, St. Clair, Benninger and colleagues use a non-invasive ultrasound approach to study changes in pancreatic blood flow during type 1 diabetes progression in mice. The authors test the novel and brilliant idea that changes in islet inflammation associated with the development of diabetes can be monitored non-invasively by measuring pancreas blood flow. The premise makes perfect sense: inflammatory conditions increase tissue perfusion and islets have a relatively large blood supply within the mouse pancreas. The results showing that pancreatic blood flow measurements can predict diabetes in different mouse models are very convincing and support the conclusions of the paper. There is a sound validation of the technique; the authors establish reproducibility and consistency and provide all the necessary controls. The changes in blood flow are impressive and indeed predict diabetes progression in mice. The discussion is thorough and addresses shortcomings as well as the future applicability of the method in human subjects. After reading the paper, the reader is left wondering why this simple but powerful approach has not been tested before. The obvious next step is to implement this technique in human beings, or at least in primate models.

There are some issues that should be addressed to make this fine manuscript even stronger.

We appreciate the reviewer indicates that our idea is “the novel and brilliant“ and that this is a “fine manuscript”.

Issue 1: The main issue is whether or not the approach transfers well to the human situation. It is encouraging that ultrasound measurements of pancreatic blood flow have already been used in human subjects using a similar technique, as referenced by the authors. It is also clear that there are substantial changes in the human islet vasculature in diabetes. What is not clear is what proportion of the pancreas blood flow readout will be representative of islet blood supply. The vasculature of the human islet is almost continuous with that of the exocrine tissue and does not show the higher vascular density typical of mouse islets (Brissova et al., 2015, J Histochem & Cytochem 63:637; Cohrs et al., 2017, Endocrinology 158:1373). If the contribution of islet blood flow to total pancreas blood flow is too small in the human pancreas, changes occurring during diabetes progression may remain undetected. This study cannot address this issue, but this potential limitation has to be addressed and discussed.

We agree this is an important point worthy of further discussion. In humans, recent studies (those the reviewer highlight, as well as S-C Tang et al Diabetologia 2017; Shah et al PLoS ONE 2016) show important differences between mouse and human islet vasculature. Human islets show a reduction in vessel density and a small decrease in vessel diameter compared to mouse [Cohers et al, Brissova et al, Tang et al], although vessel density was greater in the younger individual (19yr) examined, which is closer to the age range of T1D onset [Cohers et al]. Human islets also showed differing architecture of islet arteriole feeding, as well as microvasculature that shows more integration with the surrounding exocrine [Tang et al]. However, we do note that mouse and human islets both show changes in T2D [Brissova et al, Shah et al]. While examination in pre-T1D in humans has not to date been published, we are aware of a study currently in review [Martha Campbell-Thompson, personal communication] which shows a decrease in islet vessel diameter but increase in density,

with no change in exocrine vessel diameter, in established T1D when insulinitis is substantially less than in pre-symptomatic T1D. However, there are potential limitations such as neuroendocrine-tumor derived factors in the study by Cohers et al, the subject age (generally >40 years in all studies), with an age dependence observed by by Cohers et al.

The reduced volume of vasculature taken up by the islet may reduce the link between islet vascular changes (reflecting disease) and total pancreas blood flow measurement. However, the less localized microvasculature in human (Tang et al) has the potential to increase the link in human since a greater region of vasculature would potentially respond to islet infiltration (this may explain reported incidences of exocrine infiltration). However, we appreciate these are speculative statements.

Nevertheless we include in the discussion that there may be differences in human in the way pancreas blood flow measurements reflect islet vascular changes and blood flow changes; and that there are multiple differences between mouse and human microvascular organization – see Discussion page 13 paragraph 1 (2nd line).

Issue 2: It is known that anesthetics including isoflurane can change glycemia by reducing insulin secretion. Was this controlled for in the measurements? Because not all animals respond equally with changes in glycemia to anesthesia, this may account partially for some of the variability.

We ensured the level of anesthesia and duration of anesthesia was consistent between all measurements, and was for a duration of 10-15 minutes (C57Bl6 animals) or 20-25 minutes (NOD, NOD;Rag1ko, NOD-scid animals). Despite this consistency we do acknowledge that some animals may respond differently to anesthesia. We did record glucose levels both before the ultrasound scan before anesthesia, and at the end of the scan while under anesthesia. We did not observe any significant correlation between glucose levels at the end of the scan and the reperfusion rate or amplitude (shown below), indicating no significant effect of the animal state under anesthesia. We also did not observe any association with the change in blood glucose and the reperfusion rate indicating no significant effect of the animal response to anesthesia. We now include the reperfusion rate data in supplemental information Figure S1. We also describe these measurements – see Results page 5 paragraph 1 (18th line).

Issue 3: It is important to use appropriate statistical tests when multiple comparisons are made. Student's t tests were used inappropriately in several of the figures (e.g. Figure 3B, C, and H).

We understand the confusion as in figure captions we only state t-tests – this is because we only compared animals in the same group between early and late in diabetes. We did not for example compare STZ-treated with control treated, or AT mice with control mice. However as discussed above with respect to reviewer 1's comment and following reviewer 3 comments, we have modified our figures such that where multiple groups are presented an ANOVA is used, but where we compare a single group at 2 time points a paired t-test is used. In each figure caption, we now indicate which comparisons utilized a t-test and which comparisons utilized an ANOVA – see Figure captions pages 25-27.

Minor points:

Line 49: Please mention the approaches that hold promise.

We state the relevant approaches now – see Introduction page 3 paragraph 1 (12th line).

Line 124, Figure 1D: how do these changes compare to what has been shown before with more invasive techniques?

These measurements show good agreement with prior measurements, particularly those using invasive optical imaging measurements under glucose clamps: upon increased glycaemia we and optical imaging measurements observed increased flow velocity and upon decreased glycemia we and prior study observed reduced flow velocity. Microsphere deposition measurements do not directly reveal velocity, providing a 'flux' measurement which is proportional to velocity, but shows consistency with optical measurements. We state this now more clearly – see Results page 5 paragraph 2 (1st line).

Line 218: What can explain the increase in vessel diameter in NOD mice before the onset of diabetes? Inflammation? What does the literature say? Some discussion is needed here.

The precise mechanisms have not been well studied in T1D. However, in models of T2D islet blood flow changes have been observed prior to changes in islet microvascular diameter [Carlsson PO et al, Am J Physiol 1998; Dai C et al Diabetes 2013]. In models of T2D, a number of factors lead to altered blood flow (pericytes, PDGF signaling, parasympathetic innervation, NO signaling). However irrespective of the contributions of each mechanism, the alerted blood flow likely leads to altered microvasculature organization. In T1D, we can thus speculate a similar situation – elevated NO that results from infiltrating T cells and inflammation will lead to increased islet blood flow and increased microvascular diameter. However in T1D it has been suggested that altered VEGF plays a role [Akirav et al Diabetes 2011], although this was speculative. We prefer to avoid being overly speculative but state that whether inflammation causes the microvascular reorganization and/or

whether the microvascular organization is a cause or consequence of altered blood flow remains to be determined – see Discussion page 13 paragraph 1 (2nd line).

Line 243: For this pharmacological intervention, please cite references using this protocol
We cite the relevant study utilizing antiCD4.

Line 251: Authors mean Figure 5E, not 5F
Thank you – now corrected.

Line 280: Stating that a “multitude” of models were used is a stretch because two and a half models were used.
Agreed - we remove this descriptor.

Reviewer #3:

St Clair et al report on the use of contrast-enhanced ultrasound (CEUS) measurement of pancreatic blood flow dynamics in mice as predictor of diabetes progression and therapeutic reversal. This reviewer finds this paper to be an exciting and important concept with significant potential for translation into humans, and hopes that the authors will be able to address the issues raised.

We thank the reviewer for finding this manuscript “to be an exciting and important concept with significant potential for translation into humans”

Strengths:

1. While there is a recent paper describing the use of ultrasound in BB rat model of diabetes (PLoS ONE 12(6): e0178641.), the use of CEUS to image pancreatic inflammation is novel and there is an unmet need to “see” what is occurring in the pancreas as a predictor of disease progression, to evaluate response to therapy aimed at beta cell preservation, and to better understand the pathophysiology of disease in human type 1 diabetes.

We thank the reviewer for this comment, and also for bring attention to another recent study in the BB rat model (albeit not showing disease prediction and reversal and measuring slightly different parameters). Nevertheless we cite this article – see Discussion page 12 paragraph 3 (10th line).

2. There has been no real success in imaging pancreatic beta cells themselves, thus the idea of imaging “inflammation”. In the current pre-clinical study, the investigators evaluate the utility of whole pancreas blood flow reperfusion rate and amplitude by CEUS. Whether the sensitivity and specificity seen in small animals can be translated to humans without employing additional imaging modalities for isolating the pancreas for transcutaneous US will need to be tested. Nonetheless, CEUS has significant advantages for future clinical evaluation and use since the lack of radiation and likely very high safety suggests it could be used for longitudinal studies as well as in pediatric age groups.

We are glad the reviewer recognized the potential. We absolutely agree it is not trivial to translate this approach to humans. However as the reviewer recognizes this approach can at least be tested in humans, its potential is supported by our study and others, and if it were to translate to humans ultrasound is a highly convenient modality that avoids limitations associated with other methods.

Weaknesses

In addition to logistical and safety considerations, key elements for translation of new tools for diagnostic or prognostic evaluation in humans depend on sensitivity and specificity, variance in

populations, predictive values, and within subject reproducibility among others. The investigators have performed multiple assessments to address many of these elements, but their conclusions would be strengthened by addressing the following items:

1. Reproducibility and variance data:

a. Reproducibility and variance are important to translation. Thus, the authors should consider moving components of the figures from supplement to the main paper.

b. (S1A-C) What is the explanation for the wide variance in the C57BL6/J mice in both measures of rate and amplitude? These are genetically identical mice who presumably have no insulinitis or abnormal glucose. Moreover, mean and range of amplitude in both NOD and NOD-Scid mice (also without insulinitis or abnormal glucose) appear quite similar. Explain.

c. (S1D,E) Reproducibility over two visits should be shown for each individual mouse. The time between the tests should be shown. A T test is not appropriate for these comparisons. Moreover, there is significant variability in response of amplitude to glucose (Fig 1D) that is not considered. Text indicates no association with “day/time, reagents or operators”, yet this is not clear from legend or methods section and no data is shown. What is “normalized” amplitude and why is that used in this figure and no other?

a: We tried to avoid over-burdening the reader with data and so included additional data in the supplemental figure. However, we agree some supplemental figures would be useful (for reasons indicated), particularly when referenced in the discussion. Thus we have added or moved additional data to Figures 2, 3, 5 to show changes by mouse. Where no significant change is observed (e.g. amplitude in STZ or AT models) we include data in supplemental figures – see Figures SS3, S5, S8.

b: With respect to the NOD and NOD-scid question, the NOD mice are solely those at 6 weeks and thus show relatively low levels of insulinitis. We would therefore expect similar mean and range of measurements for rate and amplitude, as the background of these mice is identical. Of course we recognize there is some low level insulinitis starting at 6 weeks, but these results indicate the limit of detection for disease.

With respect to the C57Bl/6 mice, we do note that there are 30 animals in figure S1A and one’s attention may be drawn to the more extreme values. The SD/mean reperfusion rate for B6 is 73% (for NOD (6w) is 68%, for NOD-scid 97%) and for B6 amplitude is 43%, which are reasonable for any experiment, especially given than the change in mean reperfusion rate in NOD mice is >400% from 6w to 12w. We now mention the B6 SD/mean - see Results page 5 paragraph 1 (14th line).

Given the mean absolute change, below in sub-point c, we would conclude that the variability is not an intrinsic difference in the mouse but is a composite of variability due to mouse physiology (which may include variations in heart rate, but does not significantly include temperature, breathing rate, mouse weight) and that of the measurement.

c: We now show the day to day variability for each mouse and include the day number for the x-axis. We appreciate a test is not necessary here and instead determine the mean absolute change (reperfusion rate ~78%, amplitude ~45%), which we note is similar to the SD of the measurements - we now mention this difference - see Results page 5 paragraph 1 (16th line).

We have removed the text indicating no association with “day/time, reagents or operators as this is a presumption – aside from day, we did not explicitly test these dependencies. Normalized amplitude is the same as ‘amplitude’ used elsewhere in the manuscript and we reword this. The signal amplitude is of recovery (video intensity) is normalized to pre-ablation signal, to factor out microbubble and system variations. We have corrected this to avoid confusion.

2. Figure 1 D, E

a. This experiment was done to determine the effect of increasing (GTT) or decreasing (ITT) glucose values on the blood flow measures. The results should be presented to make the relationship clear (i.e. each animal glucose on one axis and blood value measures on the other).

b. It would be useful to know if the animals undergoing GTT were the same as those that underwent ITT; while the baseline glucose values appear similar, there are differences in baseline rate and amplitude measures. This raises the question as to whether the differences in baseline measures

exceed the differences from GTT or ITT testing and again points to questions about variability in measures in the same animal over time.

a: We present below the reperfusion rate plotted vs glucose level. We believe this is less clear than the way presented in the main figures. However we appreciate a need to examine for a potential relationship between blood glucose change and reperfusion rate change, and thus include the below data for glucose vs rate during GTT or ITT - see Supplemental Figure S1.

b: The same animals were used for GTT and ITT. We do note the spread in baseline rates and amplitude is tighter in GTT data than ITT. Nevertheless we examined results by mouse and compared changes in glucose levels between tests and changes in reperfusion rate between tests. We found no association between the response of a mouse in the GTT and its response in the ITT, in terms of glucose changes, glucose levels, reperfusion rate changes, reperfusion rate levels, amplitude changes and amplitude levels. We present below results comparing the glucose change between each test and the reperfusion rate change between each test.

3. Figure 2

a. With the small N, it would be best to show individual data in 2A, and to connect the lines both in STZ and control animals (2B,C). It would also be useful to plot the glucose value at time of measurement of blood flow.

b. Assure that the N in the figure matches the N in the legend

c. The authors assert that STZ (50) represents animals without “overt hyperglycemia”; and that the animals are “euglycemic” except for the glucose intolerance suggesting that this represents a pre-diabetic state. There is no evidence that the animals are “euglycemic” particularly given the dramatic hyperglycemia to the GTT in many. The authors also consider that there is an increase in rate in these animals; however, this difference is clearly limited to 2 animals – which, since the authors note that glucose and blood flow are related, could be due to hyperglycemia. These data do not support the idea that the rate is increased during “pre-clinical” period.

a: We modify panel A of figure 2 as suggested to provide individual data points, and to provide additional panels (as in figure 3) to connect the individual points– see figure 2 and figure S3. We also modify the lower dose STZ panels in the same manner.

b: We thank the reviewer. The n referred to number of mice at the start, but a subset of stz-treated mice receiving the higher dose (70mg/kg) and a set of control mice did die prior to 2w post measurements (ie they did not necessarily die as a result of diabetes). These animals are excluded from the graphs showing connections between mice, but are included in the graphs showing all mice to facilitate comparison of baseline measurements. We adjust the caption to factor this – see Cations page 25, figure 2 caption. For D,E,F 'n' is accurate.

c: Ad lib glucose measurements for the mice in figure 2 e,f was not significantly different 2w after STZ delivery compared to control animals and mice prior to injection. This data is now included in supplemental figure S4 (where we summarize all glucose levels and reperfusion rate data, following reviewer1 comments). We also agree that figure 2e could be plotted better: all animals did show an increase in reperfusion rate, although in some animals this was huge and some it was small. We now utilize the same scale as that of b and also present data as suggested in a, so as to better see these changes – see figure 2g.

4. Figure 3

a. *Figure 3 consists of analysis in NOD and NOD Rag1-/- mice to illustrate the relationship of blood flow dynamics with disease progression. As evident from 3A, NOD animals develop hyperglycemia at different time points – with some without progression. As such, presenting the group data at selected time points (3B, C) is not particularly helpful. More useful is 3D – showing individual changes over time; however, this plot should also include 18 week data and a similar plot should be done for the amplitude data since the authors suggest elsewhere that an increase and rate and decrease in amplitude both occur during disease progression. Importantly however, since glucose and blood flow are related, showing individual animal blood flow data over time along with the glucose over time would allow the reader to better understand the relationship of blood flow measures with disease progression (at least as measured by these random glucose values). Similarly, the group data (3E,F) are not as useful as the longitudinal data (3G), which should include the 18 week time point.*

b. *The authors use the NOD Rag1-/- mice as controls. While these animals do not develop diabetes, another control to strengthen the authors' conclusions, with an intact immune system, to consider are NOD male animals or showing longitudinal data in the C57BL6/J mice to provide information about how these variables change with age over time in health.*

c. *3H depicts “steady-state” intensity for which, while noted in methods, there is no description in the legend. It is not clear why this is an important measure. Are the authors suggesting that “steady-state” analysis provides additional information to the dynamic measures that have predictive or mechanistic value? Moreover, this graph as well as 3K present bar plots and means when individual longitudinal data would be more informative. Separately, the N for 3H is not provided in the legend. If same as for other figures, there are 23 NOD and 6 NOD Rag1-/-. From the lack of the asterisk, the authors imply no changes in this measure over time in the NOD Rag1-/- (also line 182); but the small number of animals makes this an uncertain interpretation.*

d. *Figures 3I, J uses all data points from 23 mice (62 scans). Thus, these are repeated measures. A different analytic approach should be used addressing this issue and the graphs should indicate the individual mice.*

e. *Figure 3K, data better shown as individual dots rather than bar graphs. Moreover, figure legend indicates the parameters were those that progressed to diabetes < and > 10 weeks, while the figure itself uses > and <5 weeks. In neither case is the = clear. There is no explanation as to why 5 and 10 weeks were chosen.*

a: We appreciate presenting data to indicate measurements in the same mouse to indicate changes is useful and thus move relevant panels from supplemental information to show this (Figure 3d,f,h and Figure S5b). We also include 18 week data in such graphs: please note since only a subset of animals were monitored at 18 weeks (those which develop diabetes later), this group is examined

separately when assessing changes to 18 weeks (as the 18 week data represents a biased sub set of all NOD mice). We also note in this instance all animals were euglycemic at the time of measurement (Figure 3b), so glucose does not vary significantly (see also response to reviewer 1 comment 3).

We also appreciate the reviewer's point that animals get diabetes at different points, and that is why in later panels we group data by time to diabetes rather than animal age. But given that data is often grouped by mouse age we believe this is also useful to present. However we do thank the reviewer for drawing our attention to the importance of changes in reperfusion: we examined how the change in reperfusion rate correlates with diabetes, and observed a significant trend where those that show a large change in rate or amplitude from 6-12 weeks are more likely to develop diabetes sooner. Changes from 12-18 weeks was not predictive, but this is not surprising as analysis at 18 weeks is on a less heterogeneous subset of mice.

Thus we have reorganized Figure 3 and supplemental Figure S5 (which used to be S3) to both include changes in rate and amplitude at each time point, and how the change in rate or amplitude correlates with time to diabetes.

b: We analyzed C57BLKS as controls for db/db animals (see response to reviewer 1 comment 3) and between 4 and 12 weeks we did not see a significant change – see below.

c: Steady state signal can be related to the volume of filling of the pancreas, albeit with less precision than the amplitude measurements. We have now moved this to the supplemental information following changes in figure 3. We also present individual points for this panel and others in Figure S5, rather than solely mean \pm s.e.m.

d: This is correct that 2-3 measures per mouse are included, so strictly our n is less. We have modified our analysis as a result – see Figure 3J and Figure S3E. As the time to diabetes and measurement for each scan in a mouse varies widely we do not think it best to average points. If we were to connect individual points the graph would be very challenging to visualize (the lines connecting everywhere).

e: We now show panel k (now figure 3K-N) as individual points. We apologize for the error in stating 10w rather than 5w. We examined both 5 and 10 weeks (both originally summarized in table 1, since removed as per reviewer 1) but present solely 5 weeks. We chose 5 weeks as this represents the largest bin that avoids overlap between separate ultrasound measurements (ie 6 weeks apart). 10 weeks is simply twice that and represents a value close to the median value of the time to diabetes from the ultrasound measurement. However the choice of 5 weeks is also consistent with prior studies [Turvey et al, JCI 2005] where 5w appeared to separate measurements. We also only chose these 2 cut-offs (5w, 10w) to avoid large numbers of multiple comparisons. However upon removing table 1 we only present the 5w cut off values.

Above: C57BLKS measurements for reperfusion rate, amplitude at 4,12 weeks age

5. Figure 4

a. The authors group data from a total of 49 islets from 10 mice. This is an insufficient number of islets examined to make conclusions about changes in vessels from 5 to 10 weeks. Moreover, as the authors illustrate previously, there is wide variability in disease progression in individual animals at these time points. The vascular measures from each animal at the time points should be associated with the CEUS measures from those animals at those time points.

We chose these numbers based on prior studies that assessed islet vascular morphology in models of T2D and insulin resistance, from which we utilized the tomato lectin procedure [Dai et al Diabetes (2013)]. In that study, the authors examined vascular area with at least “n = 4–5 mice and >100 islets/genotype” (Fig.2). The authors examined vascular diameter with “20 capillaries ... n = 3 islets/group” (Fig.1). Thus we appreciate that while we assess diameter using much greater islet numbers than previous studies, our islet count for vascular area is fewer, although it does show statistical significance. The n=49 refers actually to islets at 5 weeks, at 10 weeks we actually slightly more islets, although fewer were suitable to quantify vessel diameter. We include the precise islet numbers for every measurement. Nevertheless we have imaged additional islets and now include those, as well as displaying the data by mouse.

While reperfusion rate at 10-12 weeks was variable, the change from 6-12 weeks was less. Our assessment of vasculature between mice was also less variable (SD/mean ~16%, ~20% for coverage at 5w, 10w; SD/mean ~6%, ~9% for diameter at 5w, 10w). Unfortunately we were unable to measure reperfusion in the animals in which we assessed vasculature morphology, and so do not possess data in which to make this comparison. We would estimate however that the number of animals needed for such a correlation would be prohibitive (e.g. for correlating time to diabetes with change in reperfusion, >20 mice were used).

We do appreciate that the reviewer may still not think that the number of islets examined is sufficient (despite showing statistical significance). Thus we also examined vascular diameter in a second model using intra-vital imaging. In this case we follow a procedure also used by Dai et al by imaging islet vasculature following rhodamine-dextran infusion. In islets that we identify as having significant vascular coverage, the mean diameter of the islet increased with NOD mouse age – see below. We do not however wish to include this data within the manuscript as we already believe the existing data is sufficient. However if the reviewer strongly believes it would be worth also including to support existing measurements then we are happy to do so.

Above: Representative images of labelled microvasculature and labelled infiltrating T-cells collected via intra-vital imaging (left) and Correlations (with 95% CI) of mean vascular diameter in islet with NOD mouse age (right).

6. Figure 5

a. It is not clear why the authors use anti-CD4 in the AT model as the treatment to demonstrate effects of therapy when there is literature using anti-CD3 in NOD mice and humans.

b. 5A-C. The legend indicates 11 AT mice and 10 controls; however, the number of dots plotted, particularly for controls is less. As before, connecting the dots to demonstrate individual animals over time would be helpful, as well as associating the glucose results with the values since there is a large amount of variability in rate at 2 and 4 weeks in the AT animals.

c. 5D-F. With only a total of 14 mice – individual data would be useful rather than mean changes.

Moreover, the definition of non-responder are those that became hyperglycemic; it appears as if 1 or 2

of the non-responders may have been hyperglycemic at the time of infusion and thus, their data may not reflect (non-)response to therapy. Moreover, interpretation of this data is highly dependent upon day to day reproducibility without interventions, which as noted above (comment 1A), is not well presented.

a: We utilized antiCD4 as we were aware of this providing partial protection against protection against diabetes progression in the AT model to allow examination of responders and non-responders (we are unaware of antiCD3 having been used in this model to provide partial protection). We plan to examine the effect of other immunotherapies such as antiCD3 in future studies. We highlight this more clearly in the results – see Results page 9 paragraph 2 (4th line).

b: We have corrected the number of animals as fewer controls were used. While we did examine 10 'control animals' some were part of the antiCD4 study, and a small number of the reperfusion measurement did not work and were omitted – see Figure 5 caption.

As we already had for NOD data, and added for STZ data, we also include panels connecting the dots for AT and control animals (Figure 5C and Figure S8D).

As addressing the above question, we include association between blood glucose and NOD-scid mice in Figure S1 that indicates no association. We also summarize, as in responding to reviewer 1 comment 3, the blood glucose for all cases and the average reperfusion rate in figure S4.

c: We now also include panels showing individual data points for responder and non-responder – we originally presented it this way to test for significant differences between responder and non-responder – see Figure 5G and Figure S8E.

We do note the 2 animals that showed slight increases in blood glucose at the 2 week time point – one these was a responders and one a non-responders.

We agree that day to day variability (in either subject physiology or measurement) will ultimately limit separating responders from non-responders. However it is clear that if basing responder on 2w-4w trajectory, the majority of responders would be correctly identified, with 1 non-responder mis-identified as a responder; whereas if 4w level is used then all non-responders could be identified with potentially 3 responders misidentified as non-responders. In each case the majority are correctly identified which is far better than current approaches in clinical trials, or even in other pre-clinical studies. We emphasize this level of analysis and rigor has not been applied to prior published studies, particularly to separate responders and non-responders, but agree this level of rigor is warranted.

7. Figure 6

a. This illustration is not needed for the paper. Moreover, it suggests that disease progression is associated with increased rate and decreased perfusion; yet this is not consistent with all of the data shown. For example, no differences in amplitude in AT mice at week 4 despite many developing diabetes (Figure 5C); an inconsistent amplitude response to GTT (1D), no effect on amplitude in STZ treated animals (2C,F).

We would kindly request to retain this figure as we think it nicely summarizes our results for the non-expert. However we do certainly appreciate the amplitude is only changed significantly in NOD. Thus we omit stating amplitude and focus solely on rate (see Figure 6).

8. Table 1

a. This table aims to illustrate a predictive value for development of diabetes. As noted above, there is no rationale given for the 5 and 10 week time periods (and the categories exclude animals who develop diabetes at 5 and 10 weeks). There is no information provided as to how the value of >0.8 was derived. The analysis gives no information as to the number of measurements in each animal, nor the variance in the measures. If the authors believe that there is a rate value that predicts time to diabetes, it should be a rate value at a given time and not a group of measures over time. For

example, what is the positive predictive value of a rate of X for development of diabetes? The authors should consider use of ROC curves and models that predict categorical yes/no diabetes as well as time to diabetes. Alternatively, the authors could present the data (values) over time in those who do and do not progress so that the variability in this can be readily observed in all animals.

We appreciate the reviewers points here, specifically implementing ROC curves. However, following comments from reviewer 1 we now omit this table. Future work will utilize a prospective analysis to predict the success/failure of other immunotherapies.

9. Figure S4 (A-C)

a. Legend should clarify what “weighted” average means and, since only 5 mice whether there were differences in the number of islets measured in each mouse (legend states “minimum of 60 islets”). The title of the figure indicates that the insulinitis “correlates” with reperfusion rate; yet the data is not presented that way. The data should illustrate the relationship of the individual animal’s insulinitis score with their reperfusion rate at 12 weeks. Moreover, the legend says that S4C is the amplitude and the figure label says this is the rate. Both variables should be shown.

We have reworded the figure title (see Figure S6), as no correlation was observed over the 5 animals examined. We mainly intend to show the level of insulinitis as the time point measured. The weighted average referred to simply the mean insulinitis score- we have reworded accordingly. We also correct the mistake referring to amplitude rather than rate, and include amplitude data.

10. Other comments

a. Discussion; lines 304-306. Authors suggest that two independent parameters would be more specific and predictive. Yet, table 1 only considers one parameter.

b. Discussion: lines 323-325. The authors could use their data to test this statement.

c. Discussion: a recent publication (Roberts FR et al. PLoS ONE 12(6): e0178641) describes the use of US in BB rat and should be considered in the discussion.

As discussed above we omit table 1 now as a prospective study will be needed to test this. We also reword to indicate amplitude may provide increased specificity – See Discussion page 11 paragraph 1 (line 2).

By small number of snapshots, we refer to our current analysis, as compared to weekly measurements. As discussed above we did examine how the 6-12 week change in reperfusion rate and amplitude compared to the 12-18 week change in predicting diabetes onset, and observe much better prediction using the 6-12 week change – see Figure S5. This is likely as a result of there being greater heterogeneity in the development of diabetes at this time point, allowing greater separation between early and late diabetes progressors, whereas 18 weeks there are mostly late progressors examined which cannot readily be sub-divided.

We thank the reviewer for this suggestion and now cite this publication – see discussion page 12 paragraph 3 (10th line).

Reviewers' comments:

Reviewer #1 (Remarks to the Author):

The authors have comprehensively revised the manuscript in light of my comments and suggestions. This pre clinical work is a important contribution to the field and, I look forward following its development in the future.

Reviewer #2 (Remarks to the Author):

The authors have satisfactorily addressed this reviewer's critique. This study makes a good case for testing if ultrasound measurements allow monitoring diabetes progression in human subjects.

Reviewer #3 (Remarks to the Author):

The authors have provided additional information and clarification in response to previous comments. This remains a novel and exciting new tool that holds promise for filling an important scientific gap and due to safety, is likely to be able to be tested in individuals with or at risk for T1D. The authors have also done a large number of experiments.

However, several of the most important conclusions of the paper are not well supported. These include:

1. The conclusion that the method can be used to predict responders to therapy. The authors state that since there is a "statistical" difference between responding (n=9) and non-responding (n=3) animals in reperfusion rate (figure 5f), the rate at 4 weeks is predictive of this state. However, examination of figure 5g demonstrates the overreach in this statement. There is only one mouse whose reperfusion rate is higher than the rest. Otherwise, there is no difference between the groups. Moreover, the authors show in figure S1D the significant day to day variability of this measure. While this is shown only for the C57B6 mice, the range for the reperfusion rate completely overlaps the range in the treatment model. Thus, this conclusion is not well supported by their own data.

2. The conclusion that the method predicts further progression to disease and that this is not related to glucose. The issue regarding glucose may be interpretation of the term "euglycemic". An animal without diabetes, may still have glucose intolerance. Thus, direct comparison of the mouse's glucose with their data would be useful. The conclusion regarding prediction is from data in figure 3. Figure 3a shows 75% of the animals have diabetes by 20 weeks. The glucose data for these animals at different time points is shown only as column graph and not individual points (3b). At 12 weeks, figure 3c shows a statistical difference in reperfusion rate from baseline. It is clear that this is driven by a limited number of animals. This is particularly so for figure 3F in which about 50% of animals have an increase and 50% a decrease in reperfusion amplitude.

Importantly, the authors did not address the concern raised previously about figure 3J. In this figure there are multiple scans for each mouse. The statistical test should take into account repeated measures – or, more directly, the figure should show only the data from the first scan from the individual mouse plotted to time to diabetes. This appears to be the same issues for figures 3 K-L, where there are clearly more than 23 dots.

Minor points:

1. To facilitate interpretation, the authors should consider the use of color or symbols to link individual data points for glucose with the scan data. This should be done for Figures 1d,e as well as to connect the information from 2e to 2F -2H.
2. Figures 2D and 2H should connect the dots.
3. Figure 2F is the same data as in figure 2G. It is evident that the conclusion about STZ treatment is again driven by just a few animals, indeed 50% have no change from baseline, and 50%

increase – two with dramatic increases that the investigators have not convincingly demonstrated were not associated with increased (even if not “diabetic”)glucose.

Text:

1. Statement in abstract that “assessments were well separated from control animals” not well supported (see comments above).
2. Intro
 - a. First paragraph
 - i. Sufficient b-cell mass and glucose-stimulated insulin secretion remains to regulate blood glucose” is not really correct. Dysglycemia is evident along with decreased beta cell function prior to clinical onset of disease.
 - ii. “at disease onset, most...” The word “most” applies quantitation that is unknown. Large literature exists that beta cell function at time of clinical diagnosis may overlap with healthy control individuals. The logic does not follow that “intervention will be...”. Recommend “may”.
 - iii. This reviewer does not understand the new sentence “do not reflect the potentially to therapeutically reverse the disease”
 - iv. While this reviewer agrees that there is a need for more tools and we cannot currently assess insulins or mass, there is a large body of literature in which beta cell function is measured and used for these purposes.
3. Results:
 - a. Pages 6,7: issues with definition of “euglycemia”
 - b. Page 8 : if the authors wish to relate scan data with vascular data (figure 4), it would be useful to show the scan data vs the vascular data.
4. Discussion
 - a. There is an extensive literature that antibodies are “diagnostic” of pre-clinical T1D in humans. Beta cell decline is measured routinely by assessment of function.
 - b. Many of the conclusions not well supported and thus the discussion is overreach.

Reviewer #4 (expert in ultrasound) (Remarks to the Author):

Methods: It is not clear why microbubble destruction-replenishment was used for this study? Why not inject the microbubbles and simply capture tissue perfusion? Given the half-life of these agents is on the order of minutes, adding this delay in imaging can serious impact any time-intensity curve measurement. Why was data only collected for 10 seconds? Clinically, wash-out rate has shown to be a very promising metric for differentiating healthy from diseased tissue. Was this parameter considered or used?

Page 5, paragraph 1: The authors use the term “ablation”, which to me implies a surgical procedure. A more accurate wording would be microbubble, “flash destruction,” or, “destruction-replenishment imaging.”

Page 15, paragraph 3: The authors cite use of, “... transmit power 10%”. The convention is to report the peak pressure (in MPa) or mechanical index (MI). Also, what is a “Gate 6”?

Page 15, paragraph 4: It is not clear why a 3 to 4-micron size-isolated microbubble was used. Given a nonlinear imaging mode was used along with a high transmit frequency (i.e., 18 MHz), a smaller microbubble would theoretically produce a larger nonlinear signal, thus improving the ultrasound imaging results. Also, the bolus injected was rather low compared to what is typically used in small

animal studies. What was the rationale behind these decisions?

Page 16, paragraph 1: "Each reperfusion time-course was normalized to a 0.5s average of the steady

state NL contrast intensity immediately prior to ablation. Was this background subtraction? Why only 13

frames? Why an exponential model? Figure 1 has a clear offset which suggests no background subtraction was performed. If you properly account for the background signal, then correct equation

would be: $F(t) = A(1 - e^{-kt})$. Note the correct variable is t and not x .

Figure 1: There appears to be considerable amplitude offset on the time-intensity curves. Was this removed, say via baseline subtraction, prior to quantification? If not, this could have a serious impact on

all time-intensity curves measurements. What is the red line superimposed on the time-intensity curves? You cite use of an exponential curve fit but that is clearly not an exponential line

Below follows each of the reviewers comments (*italics*) and our detailed response to each comment, together with a description of where and how we have modified the manuscript. Where relevant we include numbers for specific reviewer comments to refer to later on. All changes in the manuscript are indicated in **red** and we underline the position in the manuscript we make relevant changes.

Reviewer #1:

The authors have comprehensively revised the manuscript in light of my comments and suggestions. This pre clinical work is a important contribution to the field and, I look forward following its development in the future.

Reviewer #2:

The authors have satisfactorily addressed this reviewer's critique. This study makes a good case for testing if ultrasound measurements allow monitoring diabetes progression in human subjects.

Reviewer #3:

The authors have provided additional information and clarification in response to previous comments. This remains a novel and exciting new tool that holds promise for filling an important scientific gap and due to safety, is likely to be able to be tested in individuals with or at risk for T1D. The authors have also done a large number of experiments.

However, several of the most important conclusions of the paper are not well supported. These include:

1. The conclusion that the method can be used to predict responders to therapy. The authors state that since there is a "statistical" difference between responding (n=9) and non-responding (n=3) animals in reperfusion rate (figure 5f), the rate at 4 weeks is predictive of this state. However, examination of figure 5g demonstrates the overreach in this statement. There is only one mouse whose reperfusion rate is higher than the rest. Otherwise, there is no difference between the groups. Moreover, the authors show in figure S1D the significant day to day variability of this measure. While this is shown only for the C57B6 mice, the range for the reperfusion rate completely overlaps the range in the treatment model. Thus, this conclusion is not well supported by their own data.

We agree that the prediction of disease progression or reversal would benefit from further support, beyond showing significant changes in reperfusion measurements associated with diabetes progression or reversal. We do note that in figure 5g, 2/3 'non-responder' measurements are separated from the responders, with the 1/3 'non-responder' overlapping with only 3/9 'responder' measurements. Further the 'non-responders' closely follow untreated AT mice, with the 'responders' showing similar separation from untreated AT mice. However we now provide statistical support for these statements. We performed ROC (receiver operating characteristic) analysis of data in figure 5 (adoptive transfer mice) to assess the ability of CEUS measurements to predict disease progression/reversal. We generated ROC curves based on control and untreated AT groups, and performed maximum likelihood analysis to determine the optimal threshold that separates the disease groups from non-disease groups. With this threshold we then performed an unbiased assessment of disease progression among antiCD4 treated AT mice and tested whether mice diagnosed as disease negative based on CEUS measurement would progress to diabetes with reduced incidence than mice diagnosed as disease positive or untreated AT mice. This data is summarized below (for the top 2 identified thresholds), where antiCD4-treated AT mice predicted to be disease negative progress to diabetes with reduced incidence compared to antiCD4-treated AT mice predicted to be disease positive, and compared to untreated AT mice.

Survival curves showing progression to diabetes of antiCD4-treated AT mice that are either predicted to develop diabetes or not develop diabetes. Prediction is based on the change in reperfusion rate measured between 2 and 4 weeks post splenocyte transfer, and whether this rate change falls above (Disease +) or below (Disease -) a reperfusion rate threshold determined to optimally separate control and AT mice.

Left: $p=0.09$ (antiCD4 Disease+ vs antiCD4 Disease-), $p=0.06$ (antiCD4 Disease+ vs AT untreated), $p=0.06$ (antiCD4 Disease- vs AT untreated). Right: $p=0.05$ (antiCD4 Disease+ vs antiCD4 Disease-), $p=0.1$ (antiCD4 Disease+ vs AT untreated), $p=0.05$ (antiCD4 Disease- vs AT untreated).

We performed similar analysis of data in figure 3 (NOD mice), generating ROC curves (based on NOD;Rag1ko and NOD groups measured at 12 weeks), performed maximum likelihood analysis, and assessed disease progression among NOD mice. We tested whether mice diagnosed as disease negative based on CEUS measurement would progress to diabetes more slowly than mice diagnosed as disease positive. This data is also summarized below, where NOD mice predicted to develop diabetes slowly progress to diabetes with significantly longer duration compared to NOD mice predicted to develop diabetes rapidly.

Survival curves showing progression to diabetes of NOD mice that are predicted to develop diabetes rapidly or develop diabetes slowly. Prediction is based on the change in reperfusion rate or change in reperfusion amplitude measured between 6 and 12 weeks and whether this rate change falls above (Disease +) or below (Disease -) a reperfusion rate or amplitude threshold determined to optimally separate NOD and NOD;Rag1ko mice. Left: $p < 0.0001$ (Disease+ vs Disease -, change in reperfusion rate). Right: $p = 0.003$ (Disease+ vs Disease -, change in reperfusion amplitude).

In response to this comment, we now include the results from this ROC analysis for antiCD4-treated AT mice, and for NOD mice in supplemental information and modify the results to describe this information. See Supplemental information S7, Results page 8 paragraph1, Results page 10 paragraph 2, Discussion page 11 paragraph 1, and Methods page 19 paragraph 2.

2. The conclusion that the method predicts further progression to disease and that this is not related to glucose. The issue regarding glucose may be interpretation of the term “euglycemic”. An animal without diabetes, may still have glucose intolerance. Thus, direct comparison of the mouse’s glucose with their data would be useful. We agree with this point.

Our data indicates that ad lib or fasting glucose levels are not associated with the CEUS measurements. However this does not address whether glucose intolerance is associated with the CEUS measurements. We can address this in two ways. Firstly, for STZ data we correlated the reperfusion rate with either the area under the curve (AUC) or the glucose at time point 120min. We did not observe any correlation between perfusion rate (nor with amplitude) with GTT AUC or 120min measurement, indicating a lack of association between glucose intolerance and CEUS measurement. We now include this data in supplemental information. See Supplemental information S3, Results page 6 paragraph 4.

Correlation between reperfusion rate and GTT blood glucose at 120 min. (left) or area under the curve (right). Displayed are control (black) and 50mg STZ data (red), with linear regression performed for STZ data points.

Secondly we performed glucose tolerance tests in NOD mice and NOD;Rag1ko mice at 6 and 12 weeks. At each time point we observed only minor differences in AUC or glucose at 120min in NOD mice compared to NOD;Rag1ko mice, with the glucose tolerance being far better than that in 50mg STZ mice. Thus glucose tolerance does not follow the trajectory of the CEUS measurement, which further indicates that changes in reperfusion rate in NOD mice are not associated with glucose intolerance. Rather than include this data we cite a prior study that shows lack of glucose intolerance in NOD mice at 12 weeks and earlier time points (Ize-Ludlow, D. et al. Diabetes 2011, citation 51) - cited Discussion page 13 paragraph 2.

[Redacted]

The conclusion regarding prediction is from data in figure 3. Figure 3a shows 75% of the animals have diabetes by 20 weeks. The glucose data for these animals at different time points is shown only as column graph and not individual points (3b). At 12 weeks, figure 3c shows a statistical difference in reperfusion rate from baseline. It is clear that this is driven by a limited number of animals. This is particularly so for figure 3F in which about 50% of animals have an increase and 50% a decrease in reperfusion amplitude. Importantly, the authors did not address the concern raised previously about figure 3J. In this figure there are multiple scans for each mouse. The statistical test should take into account repeated measures – or, more directly, the figure should show only the data from the first scan from the individual mouse plotted to time to diabetes. This appears to be the same issues for figures 3 K-L, where there are clearly more than 23 dots.

We now plot figure 3b as a dot plot to show the glucose levels – see figure 3.

With respect to figure 3c, we would contend that the significantly increased reperfusion rate is not driven by a limited set of animals – this can be seen by replotting the data on a tighter scale, which reveals the vast majority of measurements in NOD at 12weeks to be separated from the baseline 6 week data points, as well as from the NOD;Rag1ko 12 week data points. The mean and spread is also similar to that of AT mice. However the higher mean value in the NOD compared to the AT mice is likely attributed to the limited numbers of mice with very high values. While some NOD mice at 12w do show values similar to baseline, we do note diabetes development is heterogeneous in NOD mice. Our data showing a correlation between reperfusion rate and time to diabetes would suggest that mice showing a rate similar to baseline measurements have low levels of disease.

Reperfusion rate measurements in NOD mice, NOD;Rag1ko and adoptive transfer mice, plotted on tighter scale.

In figure 3F, some mice indeed do not show a decrease in amplitude from 6-12 weeks, although this is a minority- 10/37 (27%) of all NOD mice measured at 6 and 12 weeks. For those mice that developed diabetes later where measurements were made at 6, 12, 18 weeks, 9/24 (37%) mice showed an increase in amplitude between 6 and 12 weeks. Of these 9 mice, 7 subsequently showed a decrease from 12 to 18 weeks. We would agree the amplitude is a less predictive measure compared to rate (see above). However the change in amplitude from 6 to 12 weeks does have a significant level of prediction (see above), with mice showing a small decrease or an increase in amplitude developing diabetes at a later time point compared to those showing a large decrease in amplitude. The correlation between change in amplitude and time to diabetes again may suggest those showing an increase in amplitude have a lower levels of disease.

We apologize for the oversight in addressing the important point regarding multiple comparisons. Indeed data in Figure 3J-L and Figure S5E (rate or amplitude vs time to diabetes) reflects 2-3 measurements per mouse. We appreciate that a linear regression and t-test are not suitable statistical analyses for such data given potential correlations between data points. Therefore

we have utilized a more appropriate mixed-effects model (either 'fitlme()' in MATLAB or PROC MIXED in SAS) which reveals significance in both rate and amplitude with time to diabetes. Data in figure 3M,N (and S5F-K) only reflects 1 measurement per mouse (as does the new analysis of disease prediction where we generate the survival curves discussed above). As such we have updated the analysis of this data – see Figures 3J-L and Methods page 19 paragraph 1.

Minor points:

1. *To facilitate interpretation, the authors should consider the use of color or symbols to link individual data points for glucose with the scan data. This should be done for Figures 1d,e as well as to connect the information from 2e to 2F -2H.*
2. *Figures 2D and 2H should connect the dots.*
3. *Figure 2F is the same data as in figure 2G. It is evident that the conclusion about STZ treatment is again driven by just a few animals, indeed 50% have no change from baseline, and 50% increase – two with dramatic increases that the investigators have not convincingly demonstrated were not associated with increased (even if not “diabetic”) glucose.*

1. We now use different symbols to denote each mouse for data in figure 1D,E. We choose not to do this in figure 2 as we believe there are too many dots to keep track of and the varying symbols may cause confusion.
2. We appreciate the need for this and include this data in supplemental figure S3. We did initially place this data in figure 2 but found it made the figure too busy.
3. Figure 2F includes additional data points compared to figure 2G, where 2G only includes that data in which mice were measured effectively for both time points. Indeed some animals show only a small change in rate but some a large change in rate.

Text:

1. *Statement in abstract that “assessments were well separated from control animals” not well supported (see comments above).*
2. *Intro*
 - a. *First paragraph*
 - i. *Sufficient b-cell mass and glucose-stimulated insulin secretion remains to regulate blood glucose” is not really correct. Dysglycemia is evident along with decreased beta cell function prior to clinical onset of disease.*
 - ii. *“at disease onset, most...” The word “most” applies quantitation that is unknown. Large literature exists that beta cell function at time of clinical diagnosis may overlap with healthy control individuals. The logic does not follow that “intervention will be...”. Recommend “may”.*
 - iii. *This reviewer does not understand the new sentence “do not reflect the potentially to therapeutically reverse the disease”*
 - iv. *While this reviewer agrees that there is a need for more tools and we cannot currently assess insulits or mass, there is a large body of literature in which beta cell function is measured and used for these purposes.*
3. *Results:*
 - a. *Pages 6,7: issues with definition of “euglycemia”*
 - b. *Page 8 : if the authors wish to relate scan data with vascular data (figure 4), it would be useful to show the scan data vs the vascular data.*
4. *Discussion*
 - a. *There is an extensive literature that antibodies are “diagnostic” of pre-clinical T1D in humans. Beta cell decline is measured routinely by assessment of function.*
 - b. *Many of the conclusions not well supported and thus the discussion is overreach.*

1. This statement in the abstract was referring to how STZ-treated mice and controls compare; how NOD and NOD;Rag1ko compare or how control and AT mice compare, where there is strong separation. We agree the separation between antiCD4 responders and non-responders or between NOD mice progressing to diabetes less than or greater than 5 weeks is less substantial. However even in these cases, they are significant and predictive of disease. Given some overlap we instead state 'significantly separated' rather than 'well separated' to avoid confusion – see abstract page 2

2.i. we agree that there is some dysglycemia, but it is not substantial, hence an asymptomatic phase. Nevertheless we have reworded this statement to avoid the impression that a normal level of beta cell mass and insulin secretion remains – see Introduction page 3 paragraph 1.

2.ii. We understand there is not full agreement as to the level of functional beta cell mass at disease onset and thus substitute 'will' with 'may' as suggested – see Introduction page 3 paragraph 1.

2.iii. We refer to the fact that while islet-associated autoantibodies can predict T1D development, they have not been shown to reflect how effectively an individual will show disease reversal or prevention upon therapeutic treatment (responder or non responder). We have reworded this statement to avoid confusion – see Introduction page 3 paragraph 1.

2.iv. We agree a large number of indices have been employed to describe 'functional beta cell mass' (beta cell mass multiplied by beta cell function). Our results indicating changes where glucose homeostasis is intact suggests we would be able to detect signatures associated with insulinitis prior to detectable decline in functional beta cell mass. Nevertheless, we did not intend to refer to measurements of functional beta cell mass, but rather measurements of decline in total beta cell mass. We now reword this statement – see Introduction page 3 paragraph 1.

3.a. We now state euglycemia was based on ad lib glucose values, as we already state when describing NOD mouse euglycemia- see Results page 6 paragraph 4.

3.b. Unfortunately we were unable to perform longitudinal reperfusion measurements coupled with histological analysis for all NOD mice; and thus we performed histological analysis on a separate cohort of animals. However following the reviewer's comment we have altered our wording to test whether the changes in reperfusion rate/amplitude were consistent with morphological changes, rather than resulting from morphological changes – see Results page 8 paragraph 4.

4.a. We absolutely acknowledge the extensive literature regarding autoantibodies being diagnostic of preclinical T1D. However we would indicate that autoantibodies are not well suited to determine whether an individual will respond to preventative therapeutic treatment. Nevertheless we do acknowledge parts of the discussion could better reflect this and thus we indicate there are limited means, not a complete lack of means, to diagnose T1D prior to onset – see Discussion page 11 paragraph 1;

4.b. We hope the inclusion of the survival curve analysis addresses the first concern that conclusions regarding disease prediction are not well supported. We also hope the absence of a link between reperfusion parameters and dysglycemia (either hyperglycemia or glucose intolerance) addresses the second concern.

Reviewer #4:

Methods: It is not clear why microbubble destruction-replenishment was used for this study? Why not inject the microbubbles and simply capture tissue perfusion? Given the half-life of these agents is on the order of minutes, adding this delay in imaging can seriously impact any time-intensity curve measurement. Why was data only collected for 10 seconds? Clinically, wash-out rate has shown to be a very promising metric for differentiating healthy from diseased tissue. Was this parameter considered or used?

We believe rather than assessing simply tissue perfusion flux, separating the perfusion velocity and volume provides the greatest assessment of disease given their opposing changes. Indeed perfusion flux did not show robust changes with disease progression (Figure S5D).

We did assess microbubbles infusion kinetics (wash-in kinetics). However, while we saw some significant changes with disease, the changes were reduced and less consistent across models

compared to the reperfusion kinetics following microbubble destruction. This may result from infusion measurements also depending on variability in microbubble injection rate and systemic blood flow. Below we include some of our analysis of microbubble infusion. A small increase in infusion rate was observed in STZ-treated animals compared to vehicle controls, and in the AT mice compared to controls, but not in NOD mice (where the rate interestingly declined at 18 weeks) and not in antiCD4 non-responder compared to responders.

Infusion rise rate measured in STZ-treated mice (70mg or 50mg) (left) NOD and NOD;Rag1ko mice (middle); and Adoptive Transfer (AT) mice, and antiCD4-treated AT mice (separated by responder and non-responder) (right).

We also saw small increase in peak enhancement (change from baseline to peak signal) in STZ-treated, NOD mice and AT mice.

Peak enhancement (change from baseline to peak signal, normalized to baseline) measured in STZ-treated mice (70mg or 50mg) (left) NOD and NOD;Rag1ko mice (middle); and Adoptive Transfer (AT) mice, and antiCD4-treated AT mice (separated by responder and non-responder) (right).

We already included the steady state signal data for NOD mice showing a decline in signal in NOD mice. We also see a decline in STZ-treated mice and a trend in AT mice

Steady state signal measured in STZ-treated mice (70mg or 50mg) (left) and Adoptive Transfer (AT) mice, and antiCD4-treated AT mice (separated by responder and non-responder)(right).

We are happy to include the above data if the reviewer believes this necessary, but we believe this substantial amount of data may detract from the main findings of the manuscript and thus do not choose to include it.

We have also looked at the washout kinetics, considering either the washout rate, or the residual signal 5 minutes after infusion (see below). We do not observe any significant difference in STZ-treated mice (70mg) compared to B6 controls. However we do observe a trend for NOD mice to show a longer rate of decay and increased residual signal. Increased numbers may reveal this to be significant. However we choose not to include this data, as we are currently engaged in further examining changes in the washout rate and residual signal using other microbubble variants that we anticipate will show a much greater change in washout compared to these microbubbles.

[Redacted]

We only collected for at least 10s (typically 15-20s) as this time allowed for the reperfusion to largely level off to a maximum, which we attribute to the rapidly reperfusing component of the pancreas (the islets). There is sometimes a small but very slow component that we can also resolve which we attribute to a slower-reperfusing component of the pancreas (the exocrine tissue), but we have not explored this aspect as exocrine blood flow has not been well characterized and is much less relevant to diabetes diagnosis.

Page 5, paragraph 1: The authors use the term “ablation”, which to me implies a surgical procedure. A more accurate wording would be microbubble, “flash destruction,” or, “destruction-replenishment imaging.”

We have replaced ‘ablation’ with ‘flash destruction’ as suggested – see Results (page 5, paragraph 1), Methods (page 16 paragraph 1), and figure 1 caption (page 25, paragraph 1).

Page 15, paragraph 3: The authors cite use of, “... transmit power 10%”. The convention is to report the peak pressure (in MPa) or mechanical index (MI). Also, what is a “Gate 6”?

These settings are machine settings on the Vevo2100, which we state to enable reproducibility. However we are aware of convention and thank the reviewer for this prompt. Thus, we also include the mechanical index used - see Methods page 15 paragraph 3.

‘Gate 6’ is a setting on the Vevo2100 that describes a gap in acquisition to exclude motion artifacts due to breathing. However for the vast majority of data, aside from the very initial experiments, this ‘gating’ was performed manually post acquisition so we exclude this setting.

Page 15, paragraph 4: It is not clear why a 3 to 4-micron size-isolated microbubble was used. Given a nonlinear imaging mode was used along with a high transmit frequency (i.e., 18 MHz), a smaller

microbubble would theoretically produce a larger nonlinear signal, thus improving the ultrasound imaging results. Also, the bolus injected was rather low compared to what is typically used in small animal studies. What was the rationale behind these decisions?

This is an excellent point and we agree smaller e.g. 1-2um SIMBs may provide greater signal on the VEVO2100 given they will be closer to resonance. However we initially anticipated that 3-4um MBs would have a greater cross section (as cross section scales with radius ⁶) compared to 1-2um MBs and thus provide a higher signal despite being further from resonance. We did not use larger bubbles (e.g. 6-7um) as these are less stable in our hands. We also used 3-4um SIMBs as they are compatible with imaging on a clinical machine, as 1-2um MBs do not provide much contrast at clinical frequencies. We briefly mention this rationale - see Methods page 16 paragraph 1.

Regarding the bolus volume, we wished to minimize the volume injected into the animals while retaining a large signal increase. The volume used was a compromise between these two criteria.

Page 16, paragraph 1: “Each reperfusion time-course was normalized to a 0.5s average of the steady state NL contrast intensity immediately prior to ablation. Was this background subtraction? Why only 13 frames? Why an exponential model? Figure 1 has a clear offset which suggests no background subtraction was performed. If you properly account for the background signal, then correct equation would be: $F(t) = A(1 - e^{-kt})$. Note the correct variable is t and not x .

We did indeed subtract the background (single measured prior to MB infusion) as indeed this would substantially alter the measurement (Methods page 16 paragraph 2).

The “0.5s average of the steady state NL contrast intensity” that was used for normalizing time courses was background subtracted. We chose 13 frames to reflect the very end of the infusion time course prior to flash destruction. We could of course use 26 frames (1s) or more, but we reasoned this would have little impact as any noise would be mostly averaged out over 0.5s.

An exponential model was chosen both as this was used in prior literature [Rim et al, Circulation. 2001;104:2582-2587] – citation 31, and provides good quality fits to our data.

The offset in figure 1 (after accounting for background) reflects the fact that the flash destruction of the MBs is not 100%. Thus remaining MB-derived NL signal is the ‘C’ in the $F(t) = C + A(1 - e^{-kt})$. We have also corrected x for t , thank you – see Methods page 16 paragraph 2.

Figure 1: There appears to be considerable amplitude offset on the time-intensity curves. Was this removed, say via baseline subtraction, prior to quantification? If not, this could have a serious impact on all time-intensity curves measurements. What is the red line superimposed on the time-intensity curves? You cite use of an exponential curve fit but that is clearly not an exponential line

As mentioned in the above point, the background signal corresponding to that prior to MB infusion was subtracted from all measurements (Methods page 16 paragraph 2).

The red line is a moving average of the signal, not a fit – we state this now in the figure caption - see figure 1 caption page 25 paragraph 1.

REVIEWERS' COMMENTS:

Reviewer #3 (Remarks to the Author):

The authors have done additional analysis and addressed the main issues raised in previous review. The one remaining comment is the overreach in the conclusion sentence in the abstract'

AUTHORS CURRENT SENTENCE:

Thus contrast-enhanced ultrasound measurements of pancreas blood-flow dynamics provide a clinically-deployable predictive marker for disease progression in pre-symptomatic T1D and therapeutic reversal.

REVIEWER COMMENT:

While CEUS has been used clinically in various settings, we don't know whether this is "clinically-deployable" yet. The authors should consider revision to indicate that their data does suggest that his tool should be tested clinically.

Reviewer #4 (Remarks to the Author):

All comments were addressed and I have no more concerns worth noting.

Below follows each of the reviewers comments or editorial requests (*italics*) and our response to each comment. These changes are indicated in the manuscript using 'track changes'.

Reviewer #3:

The authors have done additional analysis and addressed the main issues raised in previous review. The one remaining comment is the overreach in the conclusion sentence in the abstract'

AUTHORS CURRENT SENTENCE:

Thus contrast-enhanced ultrasound measurements of pancreas blood-flow dynamics provide a clinically-deployable predictive marker for disease progression in pre-symptomatic T1D and therapeutic reversal

REVIEWER COMMENT:

While CEUS has been used clinically in various settings, we don't know whether this is "clinically-deployable" yet. The authors should consider revision to indicate that their data does suggest that his tool should be tested clinically.

We thank the reviewer for their satisfaction with our previous analysis and response. We agree with their remaining comment about the indicated sentence in the abstract. We note the editor has edited the abstract to account for this comment and we are happy with this edit. Any alternative revision, such as to indicate that our data suggests this tool should be tested clinically would make the abstract too lengthy. However making such a comment at the end of the discussion is warranted.

Reviewer #4:

All comments were addressed and I have no more concerns worth noting.

We thank the reviewer for their satisfaction with our previous response.